# PTPσ inhibitors promote hematopoietic stem cell regeneration

Yurun Zhang [1,10], Martina Roos[2,3,4,10], Heather Himburg[2], Christina M. Termini [2], Mamle Quarmyne[2], Michelle Li[2], Liman Zhao[2], Jenny Kan[2], Tiancheng Fang[2,5], Xiao Yan[2,5], Katherine Pohl[2], Emelyne Diers[6], Hyo Jin Gim[6], Robert Damoiseaux[4,5,7], Julian Whitelegge [8], William McBride[4,9], Michael E. Jung[6,7] & John P. Chute [2,3,4,9]

Receptor type protein tyrosine phosphatase-sigma (PTPσ) is primarily expressed by adult neurons and regulates neural regeneration. We recently discovered that PTPσ is also expressed by hematopoietic stem cells (HSCs). Here, we describe small molecule inhibitors of PTPσ that promote HSC regeneration in vivo. Systemic administration of the PTPσ inhibitor, DJ001, or its analog, to irradiated mice promotes HSC regeneration, accelerates hematologic recovery, and improves survival. Similarly, DJ001 administration accelerates hematologic recovery in mice treated with 5-fluorouracil chemotherapy. DJ001 displays high specificity for PTPσ and antagonizes PTPσ via unique non-competitive, allosteric binding. Mechanistically, DJ001 suppresses radiation-induced HSC apoptosis via activation of the RhoGTPase, RAC1, and induction of BCL-X$_L$. Furthermore, treatment of irradiated human HSCs with DJ001 promotes the regeneration of human HSCs capable of multilineage in vivo repopulation. These studies demonstrate the therapeutic potential of selective, small-molecule PTPσ inhibitors for human hematopoietic regeneration.

---

[1] Molecular Biology Institute, University of California, Los Angeles (UCLA), Los Angeles, CA 90095, USA. [2] Division of Hematology/Oncology, Department of Medicine, UCLA, Los Angeles, CA 90095, USA. [3] Eli and Edythe Broad Center for Regenerative Medicine and Stem Cell Research, UCLA, Los Angeles, CA 90095, USA. [4] Jonsson Comprehensive Cancer Center, UCLA, Los Angeles, CA 90095, USA. [5] Department of Molecular and Medical Pharmacology, UCLA, Los Angeles, CA 90095, USA. [6] Department of Chemistry and Biochemistry, UCLA, Los Angeles, CA 90095, USA. [7] California Nanosystems Institute, UCLA, Los Angeles, CA 90095, USA. [8] Department of Psychiatry and Behavioral Sciences, UCLA, Los Angeles, CA 90095, USA. [9] Department of Radiation Oncology, UCLA, Los Angeles, CA 90095, USA. [10] These authors contributed equally: Yurun Zhang, Martina Roos. Correspondence and requests for materials should be addressed to J.P.C. (email: jchute@mednet.ucla.edu)

Hematopoietic stem cells (HSCs) express several receptor tyrosine kinases (RTKs), including TIE2, fetal liver tyrosine kinase 3 (FLT3), and vascular endothelial growth factor receptor 1, which regulate HSC maintenance, proliferation, and differentiation[1–5]. RTK signaling is carefully balanced by the action of intracellular and receptor protein tyrosine phosphatases (PTPs), which dephosphorylate tyrosine residues on RTKs[6,7]. Small-molecule RTK inhibitors have been successfully developed for clinical practice and are widely utilized in oncology, including imatinib (BCR-ABL, PDGFR, c-KIT) and other BCR-ABL inhibitors, erlotinib, and gefitinib (epidermal growth factor receptor), and lapatinib (HER-2)[8–10]. Therapeutic targeting of PTPs in clinical practice has progressed more slowly, although the immunosuppressive agents, cyclosporine and FK506, which indirectly inhibit a phosphatase, calcineurin, provide evidence that phosphatase inhibitors can provide important clinical benefits[11]. Anti-sense oligonucleotides targeting PTP1B also have been demonstrated to lower blood glucose and increase insulin sensitivity in Type II diabetes mellitus, and have advanced in clinical trials[12]. Challenges to the successful development of PTP modulators include lack of selectivity, as the active enzymatic sites of PTPs may be conserved across the PTP family, and cell permeability, as highly charged anionic phosphate mimetics that could inhibit PTPs do not cross the cell membrane[11,13].

Genetic studies have revealed the function of two intracellular PTPs, Src-homology-2-domain containing tyrosine phosphatase 2 (SHP2) and phosphatase of regenerating liver 2 (PRL2), in regulating HSC fate[14,15]. A gain-of-function mutation in SHP2 promoted HSC cycling and increased HSC-repopulating capacity and the development of myeloproliferative disease in mice[14]. Deletion of PRL2 was associated with a decline in HSC self-renewal capacity in mice[15]. However, the functions of receptor PTPs in regulating HSC fate remain largely unknown. We recently discovered that pleiotrophin (PTN), a paracrine growth factor secreted by bone marrow (BM) stromal cells and endothelial cells (ECs), promoted HSC self-renewal and regeneration via binding and inhibition of receptor PTPζ[16–19]. PTPζ-deficient mice display myeloid proliferation, increased numbers of BM HSCs and progenitors, and increased HSC competitive repopulating capacity[17]. We subsequently discovered that murine and human HSCs differentially express another receptor PTP, PTPσ[20]. BM cells from PTPσ-deficient mice displayed significantly increased competitive repopulating capacity compared with PTPσ-expressing BM cells and human CB HSCs negatively selected for PTPσ surface expression demonstrated >10–fold increased repopulating capacity in NOD/SCID IL2 receptor-γ chain-deficient (NSG) mice[20]. Based on these results, we hypothesized that PTPσ negatively regulates HSC self-renewal, and that pharmacologic inhibition of PTPσ could therapeutically promote HSC self-renewal. Here we report the development of a small molecule that selectively inhibits PTPσ via allosteric binding in BM HSCs. Systemic administration of the PTPσ inhibitor, or its analog, promoted HSC regeneration, accelerated hematologic recovery, and improved survival in irradiated mice. Furthermore, treatment of irradiated, human HSCs with the PTPσ inhibitor promoted the regeneration of human HSCs capable of multilineage repopulation in NSG mice. These studies reveal a class of selective PTPσ inhibitors with therapeutic potential to promote human hematopoietic regeneration.

## Results

### Development of a selective and allosteric PTPσ inhibitor.
We hypothesized that PTPσ inhibition could promote HSC self-renewal and sought to develop a small molecule with specific inhibitory activity against PTPσ. We identified an organic compound, 6545075 (Chembridge), which was reported to have binding and inhibitory properties against the catalytic domain of PTPσ[21]. We searched a library of 80,000 small molecules in an attempt to find compounds with similar structural features, namely 2-arylamino-1-arylpropenones, and identified two additional compounds, 6515205 and 5483071, which we hypothesized to have the potential to bind to PTPσ (Fig. 1a). We tested the inhibitory activity of these three molecules in a direct PTPσ enzymatic assay (Fig. 1b)[22]. Compound 5483071 (3071) displayed strong inhibitory activity against PTPσ, with a half-maximal inhibitory concentration (IC50) of 1.5 μM, whereas the other compounds demonstrated lower PTPσ inhibitory activity. We then prepared a synthetic sample of 3071, named DJ001, and confirmed the inhibitory effect of DJ001 in the PTPσ enzymatic assay (Fig. 1c). Importantly, DJ001 displayed no inhibitory activity against 20 other phosphatases, with only modest inhibitory activity against Protein Phosphatase 5 (Supplementary Fig. 1a). These data suggested that DJ001 was a highly specific inhibitor of PTPσ.

We next determined that DJ001 can exist in two different stereo-isomeric forms, the (Z)- and (E)-isomers (Supplementary Fig. 2a). The (Z)-isomer displayed significantly increased binding affinity to both the PTPσ active (catalytic) site and the allosteric site compared with the (E)-isomer (Supplementary Fig. 2b, c). Importantly, the (Z)-isomer also demonstrated a 1.3 kcal/mol preference for binding to the allosteric site, located between domains 1 and 2 of PTPσ, over the catalytic site (−9.0 kcal/mol vs. −7.7 kcal/mol) (Fig. 1d). Enzyme kinetic analysis via substrate titration studies demonstrated that DJ001 functions as a non-competitive inhibitor of PTPσ, consistent with its allosteric binding properties (Fig. 1e). These studies revealed that DJ001 non-competitively inhibits PTPσ through a unique mechanism.

### PTPσ inhibition promotes hematopoietic regeneration.
In order to determine whether PTPσ inhibition could affect HSC growth during homeostasis, we cultured non-irradiated BM CD34−ckit+sca1+lin− (CD34−KSL) HSCs with media (containing 20 ng/mL Thrombopoietin, 100 ng/mL stem cell factor (SCF), 50 ng/mL Flt3 ligand, TSF) with or without 10–1,000 ng/mL DJ001 for 7 days. Treatment with DJ001 increased the percentages and numbers of BM KSL cells in culture compared with control cultures (Supplementary Fig. 3a). DJ001 treatment also significantly increased the numbers of colony forming cells (CFCs) in 3 day culture of BM KSL cells (Supplementary Fig. 3b). Systemic treatment of adult C57BL/6 mice with 5 mg/kg DJ001, subcutaneously every other day for 30 days, caused no significant changes in complete blood counts, BM HSC or progenitor cell content, or BM CFCs compared with vehicle-treated controls (Supplementary Fig. 3c–e).

In order to determine whether PTPσ inhibition could promote HSC regeneration following myelosuppressive injury, we irradiated BM KSL cells with 300 cGy and placed in TSF media with and without 1 μg/mL DJ001 for 3 days. DJ001 treatment increased recovery of BM CFCs and multipotent colony-forming unit–granulocyte erythroid monocyte megakaryocyte (CFU-GEMM) colonies compared with control cultures (Fig. 2a and Supplementary Fig. 4a). In a complementary study, we irradiated mice bearing constitutive deletion of Ptprs, the gene that encodes PTPσ (Ptprs−/− mice), with 600 cGy total body irradiation (TBI). Irradiated Ptprs−/− mice displayed increased recovery of BM CFCs at day + 10 compared with irradiated Ptprs+/+ mice (Fig. 2b). These results suggested that deletion of Ptprs or PTPσ inhibition comparably promoted hematopoietic progenitor cell regeneration following irradiation.

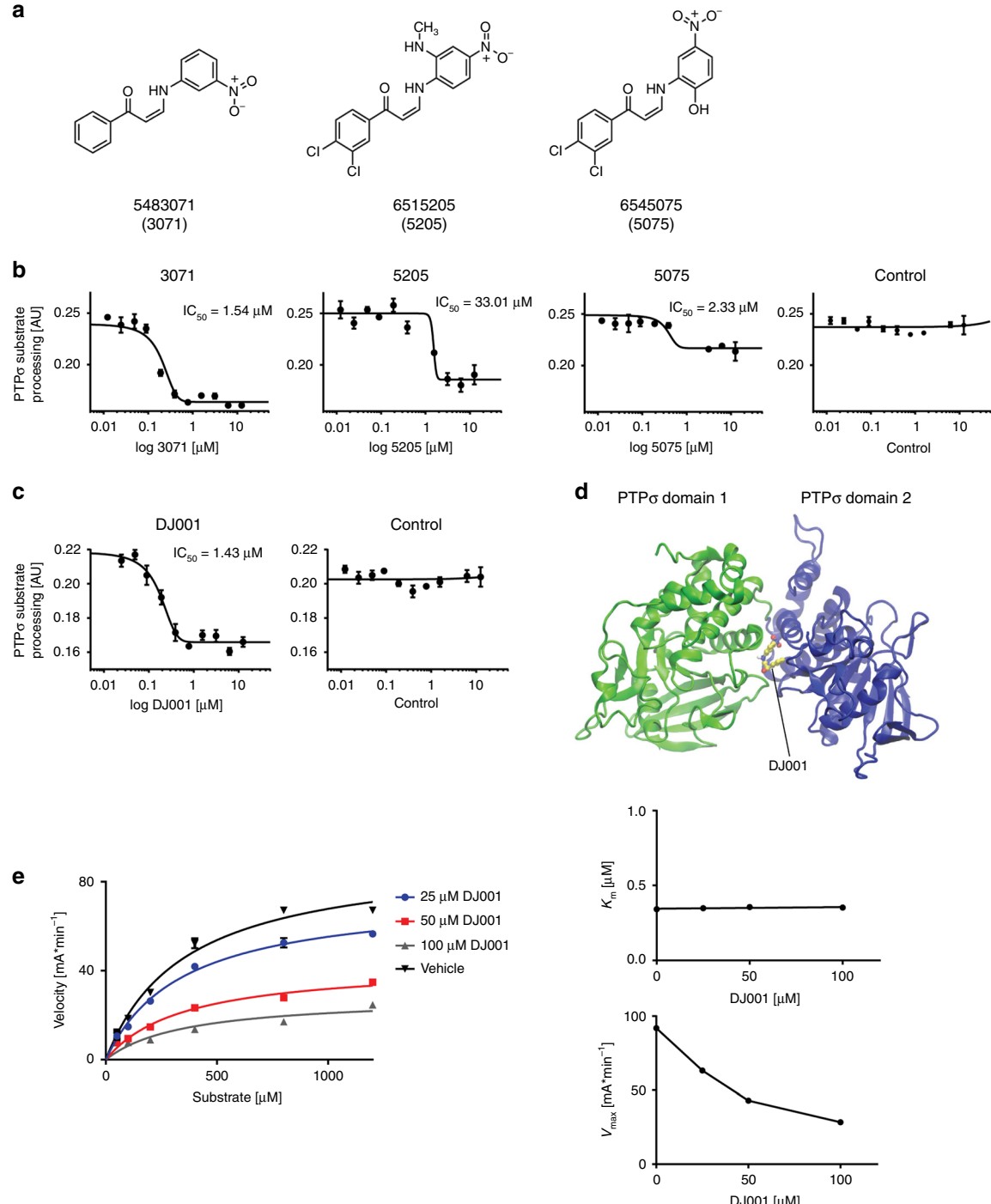

**Fig. 1** DJ001 is a non-competitive, allosteric inhibitor of PTPσ. **a** Chemical structures of compounds 3071, 5205, and 5075. **b** Concentration-inhibition curves and IC$_{50}$ values for 3071, 5205, 5075, and control (DMSO) following incubation with PTPσ ($n = 3$). **c** Concentration-inhibition curves and IC$_{50}$ values for DJ001 and control following incubation with PTPσ ($n = 3$). **d** In silico three-dimensional docking of DJ001 (Z isomer) (represented as ball and stick in yellow color) to the PTPσ allosteric binding site located between domain 1 (green) and domain 2 (blue) of PTPσ. **e** At left, substrate titration reveals DJ001 as a non-competitive inhibitor that inhibits substrate catalysis ($V_{max}$), but not substrate binding of PTPσ (constant $K_m$). At right, plots of $K_m$ and $V_{max}$ as a function of compound concentration delivered from nonlinear regression in substrate titration. Error bars represent SEM. Source data are provided as Source Data File

In order to determine whether PTPσ inhibition could promote hematopoietic regeneration in vivo, we irradiated adult C57BL/6 mice with 750 cGy TBI and treated with 5 mg/kg DJ001 or vehicle subcutaneously every day from day + 1 to day + 10. At day + 10, DJ001-treated mice displayed more than twofold increased peripheral blood (PB) white blood cells (WBCs), neutrophils, and lymphocytes compared with vehicle-treated controls (Fig. 2c). DJ001 treatment also accelerated the recovery of BM KSL cells, which are enriched for hematopoietic stem/progenitor cells (HSPCs), ckit⁺sca-1⁻lin⁻ myeloid progenitors, and CFCs following TBI (Fig. 2d, e and Supplementary Fig. 4b, c). DJ001 treatment did not alter the percentages of BM

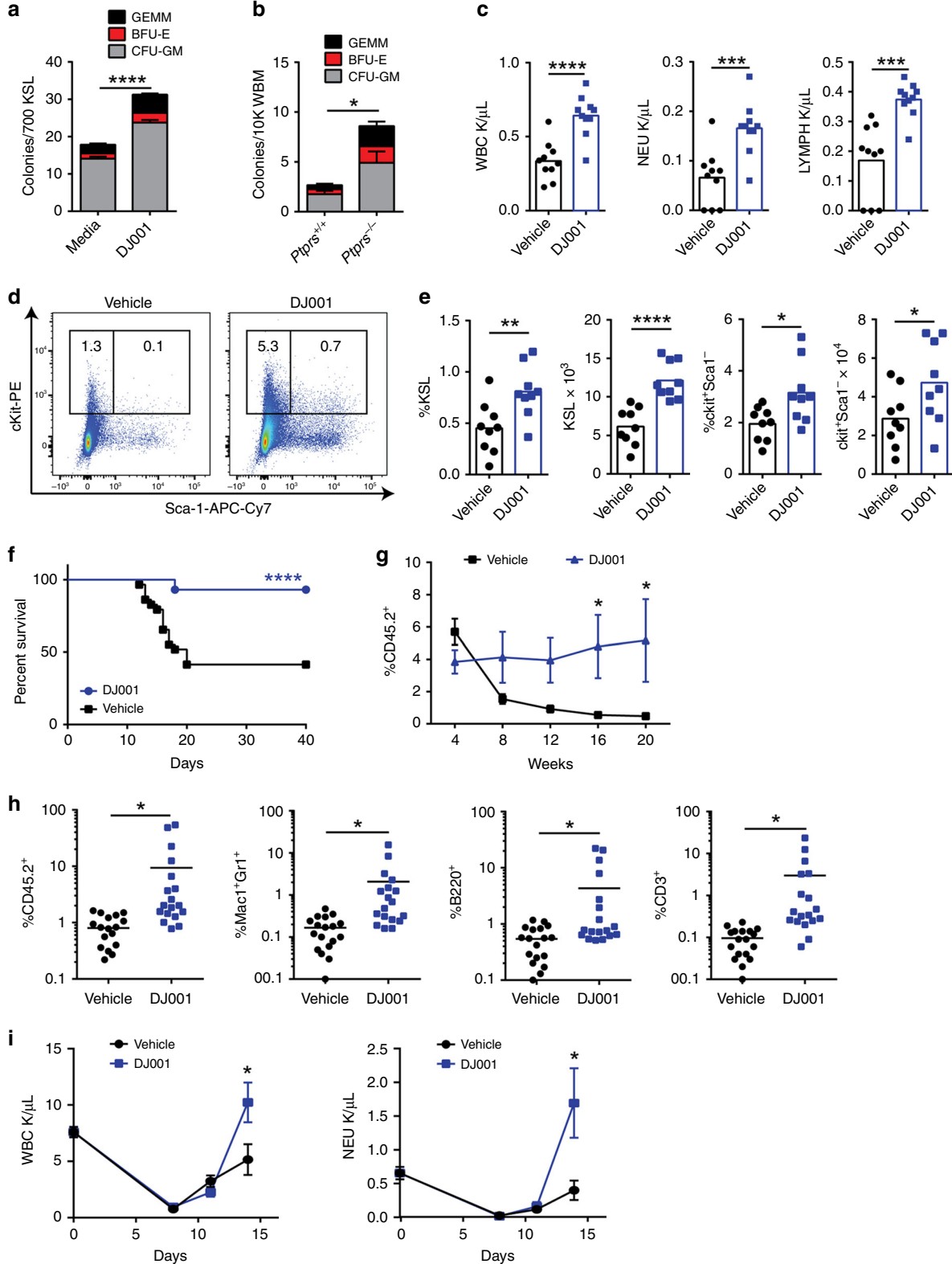

megakaryocyte erythroid progenitors (MEPs), common myeloid progenitors (CMPs), granulocyte monocyte progenitors (GMPs), or PB myeloid, B-cell or T-cell populations, but did cause a modest increase in percentages of BM common lymphoid progenitors (CLPs)(Supplementary Fig. 4d, e). Of note, pharmacokinetic (PK) analysis of adult C57BL/6 mice treated subcutaneously × 1 with 5 mg/kg DJ001 demonstrated DJ001 concentrations of 10–100 ng/mL in the PB over several hours

post injection (Supplementary Fig. 4f), a concentration range that promoted BM hematopoietic progenitor cell recovery in vitro (Supplementary Fig. 3b).

As radiation exposure can cause acute mortality via hematopoietic toxicities, we evaluated whether daily administration of DJ001 from day + 1 to day + 10 could increase the survival of mice following lethal dose TBI. Following 750 cGy TBI, 12 of 29 adult C57BL/6 mice (41%) survived to day + 40. Conversely, 27

**Fig. 2** PTPσ inhibition promotes hematopoietic regeneration. **a** Mean numbers of CFCs from BM KSL cells following 300 cGy irradiation and culture for 3 days in TSF media ± DJ001 ($n = 12$). **b** Mean numbers of BM CFCs in $Ptprs^{+/+}$ and $Ptprs^{-/-}$ mice at day + 10 following 600 cGy TBI ($n = 12$). **c** Mean numbers of PB WBCs, neutrophils (NEU), and lymphocytes (LYMPH) in mice at day + 10 following 750 cGy and treatment with DJ001 or 10% DMSO (vehicle) ($n = 10$). **d** Representative flow cytometric analysis of percentages of KSL cells and c-kit⁺sca-1⁻ progenitor cells at day + 10 post 750 cGy in the treatment groups shown. **e** Mean percentages and numbers of BM KSL cells and c-kit⁺sca-1⁻ cells at day + 10 post 750 cGy ($n = 9$). **f** Percent survival of mice after 750 cGy and treatment with DJ001 (27/29 alive) or vehicle (12/29 alive); Log-rank test. **g** Mean percentages of donor CD45.2⁺ cells in the PB of CD45.1⁺ recipient mice over time following transplantation of $5 \times 10^5$ BM cells from CD45.2⁺ mice at day + 10 following 750 cGy TBI and treatment with DJ001 or vehicle, along with $1 \times 10^5$ CD45.1⁺ BM competitor cells ($n = 18-21$/group). **h** Mean percentages of total donor CD45.2⁺ cells and CD45.2⁺ Mac1⁺Gr1⁺ myeloid cells, CD45.2⁺B220⁺ B cells, and CD45.2⁺CD3⁺ T cells in the PB of recipient mice at 20 weeks post transplantation ($n = 18$/group). **i** Mean numbers of PB WBCs and NEU counts in mice at day + 0 (pre-5FU), + 8, + 11, and + 14 following 5FU treatment ($n = 4-7$/group). Error bars represent SEM. Source data are provided as Source Data File. *$P < 0.05$, **$P < 0.01$, ***$P < 0.001$, ****$P < 0.0001$

of 29 irradiated C57BL/6 mice treated with DJ001 (93%) remained alive and healthy through day + 40 ($P < 0.0001$, Log-rank test, Fig. 2f). Therefore, systemic administration of a PTPσ inhibitor also improved survival following myeloablative irradiation.

In order to determine whether DJ001 promoted the regeneration of HSCs with in vivo repopulating capacity, we performed competitive repopulation assays using B6.SJL (CD45.1⁺) recipient mice and donor BM collected at day + 10 from C57BL/6 (CD45.2⁺) mice irradiated with 750 cGy TBI and treated daily with 5 mg/kg DJ001 or vehicle. Recipient mice transplanted with $5 \times 10^5$ BM cells from irradiated control mice, along with $1 \times 10^5$ competitor CD45.1⁺ BM cells, displayed declining donor-derived hematopoietic reconstitution over time following transplant (Fig. 2g). This is consistent with our prior observations that high-dose TBI depletes HSCs with long-term repopulating capacity[23]. In contrast, mice transplanted with the identical dose of BM cells from irradiated, DJ001-treated mice displayed significantly increased donor hematopoietic reconstitution over time compared with controls (Fig. 2g). Of note, mice transplanted with BM cells from irradiated, DJ001-treated donor mice displayed significantly increased percentages of donor CD45.2⁺Mac1⁺/Gr1⁺ myeloid cells, CD45.2⁺B220⁺ B cells, and CD45.2⁺CD3⁺ T cells in the PB at 20 weeks post transplant compared with mice transplanted with control, irradiated BM cells (Fig. 2h).

In order to determine whether systemic treatment with DJ001 could also promote hematologic recovery following myelosuppressive chemotherapy, we treated adult C57BL/6 mice intravenously with 250 mg/kg 5-fluorouracil (5FU), an antimetabolite chemotherapeutic that is widely utilized in clinical practice and causes depression of WBCs and neutrophils[24,25]. Control mice treated with 5FU displayed leukopenia and neutropenia from day + 8 through day + 14 (Fig. 2i). Mice treated with 5FU followed by 5 mg/kg DJ001 SQ daily through day + 10 demonstrated significantly increased PB WBCs and neutrophils at day + 14.

In order to validate our hypothesis that small-molecule inhibition of PTPσ could facilitate hematopoietic regeneration following myelotoxicity, we designed and synthesized more than 50 structural analogs of DJ001. One of them, DJ009, with a 3,5-difluorophenyl ring substituted for the 3-nitrophenyl unit of DJ001 (Supplementary Fig. 5a), strongly inhibited PTPσ activity in vitro (Supplementary Fig. 5b). When C57BL/6 mice were irradiated with 750 cGy TBI and subsequently treated daily with 5 mg/kg DJ009 for 10 days, we observed a significant increase in PB WBCs, neutrophils and lymphocytes, BM KSL cells, and CFCs in response to DJ009 treatment (Supplementary Fig. 5c–e). Consistent with these findings, DJ009 treatment also significantly increased survival of irradiated C57BL/6 mice (Supplementary Fig. 5f). These results provided further evidence that small-molecule PTPσ inhibitors can promote hematopoietic regeneration in vivo following myelosuppression.

**PTPσ inhibitor effects on cytokine levels**. Although our in vitro studies suggest that PTPσ inhibition directly promotes HSPC regeneration, it is also possible that treatment with DJ001 may have caused indirect effects via alterations in the production of cytokines by hematopoietic progenitor cells or their microenvironment. Indeed, hematopoietic progenitor cells have been shown to secrete several inflammatory cytokines in response to lipopolysaccharide and Pam3CSK4, a Toll-like receptor 2 ligand[26]. In order to address this, we measured 38 cytokines in cultures of non-irradiated and irradiated BM KSL cells treated with and without DJ001, and also measured the identical cytokines in the BM of non-irradiated C57BL/6 mice and irradiated C57BL/6 mice at day + 5 following treatment with DJ001 or vehicle. At 12 h following 300 cGy irradiation of BM KSL cells in vitro, we detected increased levels of lipopolysaccharide-induced CXC chemokine (LIX), macrophage inflammatory protein-1α (MIP-1α) and MIP-1β, and no other changes; DJ001 treatment decreased LIX levels following irradiation but had no effects on any other cytokines in vitro (Supplementary Fig. 6a, b). At day + 5 following 750 cGy TBI in C57BL/6 mice, we detected decreased levels of LIX and CXCL12 in the BM and increased levels of SCF and PTN. DJ001 treatment had no effects on cytokine levels in non-irradiated mice, but caused a modest increase in CXCL12 levels and a decrease in PTN levels in the BM of irradiated mice compared with irradiated, vehicle-treated mice (Supplementary Fig. 6c, d).

**PTPσ inhibition promotes HSC survival via induction of RAC1**. The mechanisms through which PTPσ regulates cellular functions are not well understood[27]. In a yeast-two-hybrid system, PTPσ was shown to dephosphorylate p250GAP, a RhoGT-Pase that attenuates RAC1 activity[27]. In the presence of PTPσ, p250GAP activity increased and RAC1 activity declined[27]. Here we tested whether PTPσ inhibition with DJ001 could alter p250GAP phosphorylation and RAC1 activation in primary hematopoietic progenitor cells. For this purpose, we developed an enzyme-linked immunosorbent sandwich enzyme-linked immunosorbent assay (ELISA) employing immunoglobulin A capture of p250GAP (Supplementary Fig. 7a). Treatment with 1 μg/mL DJ001 significantly increased phosphorylation of p250GAP in BM lin⁻ cells (Fig. 3a). Furthermore, DJ001 treatment significantly decreased co-localization of PTPσ and p250GAP in BM KSL cells, as measured by proximity ligation assay (PLA; Fig. 3b). DJ001 treatment also caused a rapid increase in RAC1-GTP levels in BM KSL cells (Fig. 3c). Importantly, DJ001 treatment did not alter the activation of other RhoGTPases, CDC42, or RhoA, in BM lin⁻ cells, suggesting that CDC42 and RhoA did not contribute to DJ001-mediated effects in hematopoietic progenitor cells (Supplementary Fig. 7b). Therefore, PTPσ repressed RAC1 activation in HSPCs via regulation of p250GAP and DJ001 promoted RAC1 activation via inhibition of PTPσ and p250GAP function. Of note, DJ001 treatment of BM KSL cells from

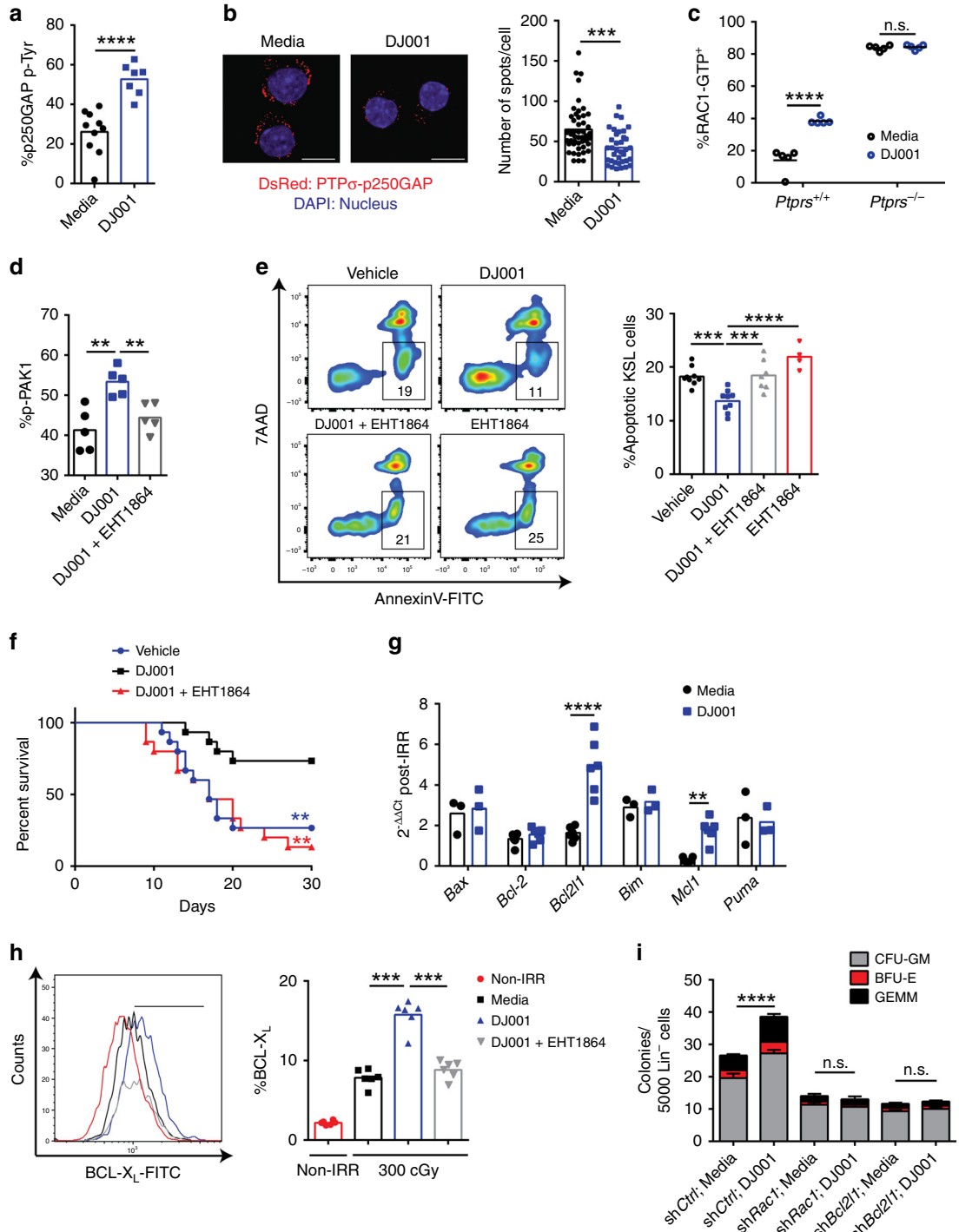

$Ptprs^{-/-}$ mice caused no change in RAC1-GTP levels, suggesting that DJ001-mediated activation of RAC1 occurred specifically via PTPσ (Fig. 3c). DJ009 also did not induce RAC1 activation in BM cells from $Ptprs^{-/-}$ mice, suggesting similar selectivity of DJ009 for PTPσ (Supplementary Fig. 7c). Treatment of BM KSL cells with DJ001 increased phosphorylation of p21-activated kinase 1 (PAK1), a substrate of RAC1, and concomitant treatment with the RAC inhibitor, EHT1864[28], abrogated DJ001-mediated phosphorylation of PAK1 (Fig. 3d).

Ionizing radiation causes HSC death via direct DNA damage, generation of reactive oxygen species, and induction of pro-apoptotic mediators[25,29]. DJ001 treatment significantly decreased the percentage of apoptotic BM KSL cells in C57BL/6 mice at 24 h

following 500 cGy TBI compared with vehicle-treated controls (Fig. 3e). Administration of the RAC inhibitor, EHT1864, abrogated DJ001-mediated anti-apoptotic effects on BM KSL cells in irradiated mice (Fig. 3e). Similarly, treatment with EHT1864 inhibited DJ001-mediated CFC generation from BM KSL cells irradiated with 300 cGy in vitro (Supplementary Fig. 7d). Systemic administration of EHT1864 also completely blocked DJ001-mediated increase in survival of irradiated mice (Fig. 3f). These results suggested that DJ001-mediated anti-apoptotic and regenerative effects on BM HSPCs were dependent on RAC pathway activation.

RhoGTPases such as RAC1 can affect multiple signaling pathways that regulate cell survival[30–34], so we next measured the

**Fig. 3** DJ001 promotes HSC regeneration via RAC1 activation and induction of BCL-X$_L$. **a** %p250GAP phospho-tyrosine (pTyr) in BM lin$^-$ cells cultured ± DJ001 (media, $n = 10$; DJ001, $n = 7$). **b** At left, representative images of PLA in BM KSL cells treated ± DJ001. Red = DsRed$^+$, PTPσ-p250GAP complex; blue = DAPI, original magnification ×63, scale bars = 10 μm). At right, numbers of DsRed$^+$ KSL cells in each condition ($n = 48$, control; $n = 36$, DJ001). **c** % RAC1-GTP$^+$ KSL cells from $Ptprs^{+/+}$ and $Ptprs^{-/-}$ mice treated ± DJ001 ($n = 5$). Two-way ANOVA with Sidak's multiple comparison test. **d** Percentages of p-PAK1$^+$ KSL cells following treatment with media alone, 1 μg/mL DJ001, or DJ001 + 6 μg/mL EHT1864 ($n = 5$). One-way ANOVA with Tukey's multiple comparison test. **e** At left, Annexin V/7AAD staining of BM KSL cells from mice at 24 h post 500 cGy and treatment with vehicle, 5 mg/kg DJ001, DJ001 + 40 mg/kg EHT1864, or EHT1864 alone. Numbers represent percentages in each gate. At right, %Annexin$^+$7AAD$^-$ BM KSL cells (media and DJ001, $n = 9$; DJ001 + EHT1864, $n = 7$; EHT1864, $n = 4$). **f** Percent survival of mice following 750 cGy and 10-day treatment with DJ001 (11/15 alive), vehicle (4/15 alive), or DJ001 + EHT1864 (2/15 alive; Log-rank test). **g** Fold changes ($2^{-\Delta\Delta Ct}$) of gene expression in BM KSL cells at 12 h post 300 cGy and treatment with or without DJ001 ($n = 3–6$/group). Two-way ANOVA with Sidak's multiple comparison test. **h** At left, BCL-X$_L$ protein levels in non-irradiated BM KSL cells (red), 300 cGy-irradiated KSL cells in media alone (black) or treated with DJ001 (blue) or DJ001 + EHT1864 (gray). At right, % BCL-X$_L$$^+$ KSL cells ($n = 6$). One-way ANOVA with Tukey's multiple comparison test. **i** CFCs from BM KSL cells at 48 h following 300 cGy and culture with and without DJ001, and with and without $shRac1$ or $shBcl2l1$ ($n = 3–9$/group). Two-way ANOVA with Sidak's multiple comparison test. Error bars represent SEM. Source data are provided as Source Data File. *$P < 0.05$, **$P < 0.01$, ***$P < 0.001$, ****$P < 0.0001$

**Table 1 Primers used for mouse gene detection**

| Target mRNA | Sequence 5′–3′ |
| --- | --- |
| Gapdh Fw | TGGATTTGGACGCATTGGTC |
| Gapdh Rv | TTGCACTGGTACGTGTTGAT |
| Bcl2l1 Fw | GACAAGGAGATGCAGGTATTGG |
| Bcl2l1 Rv | TCCCGTAGAGATCCACAAAAG |
| Bax Fw | TGAAGACAGGGGCCTTTTTG |
| Bax Rv | AATTCGCCGGAGACACTCG |
| Bcl2 Fw | TGAGTACCTGAACCGGCATCT |
| Bcl2 Rv | GCATCCCAGCCTCCGTTAT |
| Bim Fw | CGGATCGGAGACGAGTTCA |
| Bim Rv | TTCCAGCCTCGCGGTAATCA |
| Puma Fw | AGCAGCACTTAGAGTCGCC |
| Puma Rv | CCTGGGTAAGGGGAGGAGT |
| Mcl-1 Fw | AAAGGCGGCTGCATAAGTC |
| Mcl-1 Rv | TGGCGGTATAGGTCGTCCTC |
| Cdk2 Fw | CCTGCTTATCAATGCAGAGGG |
| Cdk2 Rv | TGCGGGTCACCATTTCAGC |
| Cdk4 Fw | ATGGCTGCCACTCGATATGAA |
| Cdk4 Rv | TCCTCCATTAGGAACTCTCACAC |
| Cdk6 Fw | GGCGTACCCACAGAAACCATA |
| Cdk6 Rv | AGGTAAGGGCCATCTGAAAACT |
| Cdkn1a Fw | CCTGGTGATGTCCGACCTG |
| Cdkn1a Rv | CCATGAGCGCATCGCAATC |
| Cdkn1b Fw | TCAAACGTGAGAGTGTCTAACG |
| Cdkn1b Rv | CCGGGCCGAAGAGATTTCTG |
| Cyclin E Fw | GCCAGCCTTGGGACAATAATG |
| Cyclin E Rv | CTTGCACGTTGAGTTTGGGT |
| Rac1 Fw | GAGACGGAGCTGTTGGTAAAA |
| Rac1 Rv | ATAGGCCCAGATTCACTGGTT |
| Cyclin D1 Fw | GCGTACCCTGACACCAATCTC |
| Cyclin D1 Rv | CTCCTCTTCGCACTTCTGCTC |
| Cyclin D2 Fw | GAGTGGGGAACTGGTAGTGTTG |
| Cyclin D2 Rv | CGCACAGAGCGATGAAGGT |

effect of DJ001 treatment on the expression of pro- and anti-apoptotic genes in irradiated KSL cells (Fig. 3g). DJ001 treatment had no effect on the expression of pro-apoptotic mediators, *Bim*, *Bax*, *p53 upregulated modulator of apoptosis* (*Puma*), or the anti-apoptotic gene, *Bcl2*, in BM KSL cells (Table 1). However, DJ001 treatment increased the expression of *Mcl-1* and *Bcl2 like 1* (*Bcl2l1*), which encodes the anti-apoptotic protein, BCL-X$_L$, as well as BCL-X$_L$ protein levels in irradiated BM KSL cells (Fig. 3g, h). RAC inhibition blocked the increase in BCL-X$_L$ protein levels in irradiated BM KSL cells in response to DJ001, suggesting that DJ001-mediated induction of *Bcl2l1* expression was RAC pathway dependent (Fig. 3h). In order to confirm that DJ001-mediated effects on irradiated HSPCs were dependent on RAC1 and BCL-X$_L$, we transduced BM KSL cells separately with lentiviral short

hairpin RNAs (shRNAs) targeting *Rac1* or *Bcl2l1* and measured hematopoietic progenitor cell recovery following 300 cGy irradiation. Silencing of either *Rac1* or *Bcl2l1* blocked DJ001-mediated recovery of hematopoietic progenitor cells from irradiated BM KSL cells (Fig. 3i). Taken together, these results suggested that DJ001-mediated HSPC recovery after irradiation was dependent on RAC1 and BCL-X$_L$.

RAC1 has been shown to activate ERK1/2 signaling via PAK1- and PAK2-mediated phosphorylation of MEK1 to facilitate the formation of the Raf/MEK/ERK complex[35]. Consistent with this, DJ001 treatment induced ERK1/2 phosphorylation in irradiated BM KSL cells, which was suppressed by RAC inhibition (Supplementary Fig. 7e). Furthermore, treatment with the ERK1/2 inhibitor, BVD523, blocked DJ001-mediated increase in BCL-X$_L$ protein levels in BM KSL cells and abrogated hematopoietic progenitor cell recovery following irradiation (Supplementary Fig. 7f, g). These data suggested that DJ001-mediated induction of BCL-X$_L$ in HSPCs was dependent on ERK1/2, downstream of RAC pathway activation.

**PTPσ inhibition promotes HSPC proliferation via Cdk2.** Hematologic recovery following myelosuppressive irradiation requires not only survival of long-term HSCs but also the proliferation of HSCs and progenitor cells to facilitate regeneration of essential blood elements. DJ001 treatment did not alter the cell cycle status of non-irradiated BM KSL cells in culture with thrombopoietin, SCF, and Flt3 ligand (Supplementary Fig. 7h). However, DJ001 treatment of BM KSL cells for 36 h following 300 cGy irradiation significantly increased the percentage of KSL cells in G$_2$/S/M phase compared with control, irradiated KSL cells (Fig. 4a). Evaluation of the expression of cell cycle regulatory genes revealed that DJ001 treatment did not alter the expression of cyclin-dependent kinase 4 (*Cdk4*), *Cdk6*, *Cdkn1a*, *Cdkn1b*, *Cyclin D1*, or *Cyclin D2* in irradiated BM KSL cells (Fig. 4b and Table 1). However, DJ001 treatment significantly increased the expression of *Cdk2* and *Cyclin E* in irradiated KSL cells. Transduction of BM KSL cells with *Rac1*-shRNA blocked DJ001-mediated transcription of *Cdk2* and *Cyclin E*, suggesting that DJ001-induced expression of *Cdk2* and *Cyclin E* in KSL cells was RAC1 dependent (Fig. 4c). Furthermore, treatment of irradiated BM KSL cells with SU9516, a specific CDK2 inhibitor, or the RAC inhibitor, EHT1864, suppressed DJ001-mediated induction of cell cycle progression in BM KSL cells following irradiation (Fig. 4d). These results suggested that DJ001-mediated HSC cell cycle progression following irradiation was dependent on RAC pathway activation and CDK2.

In order to test whether systemic treatment with DJ001 altered HSC proliferation in vivo following irradiation, we measured BrdU incorporation in BM KSL cells of C57BL/6 mice at day + 10

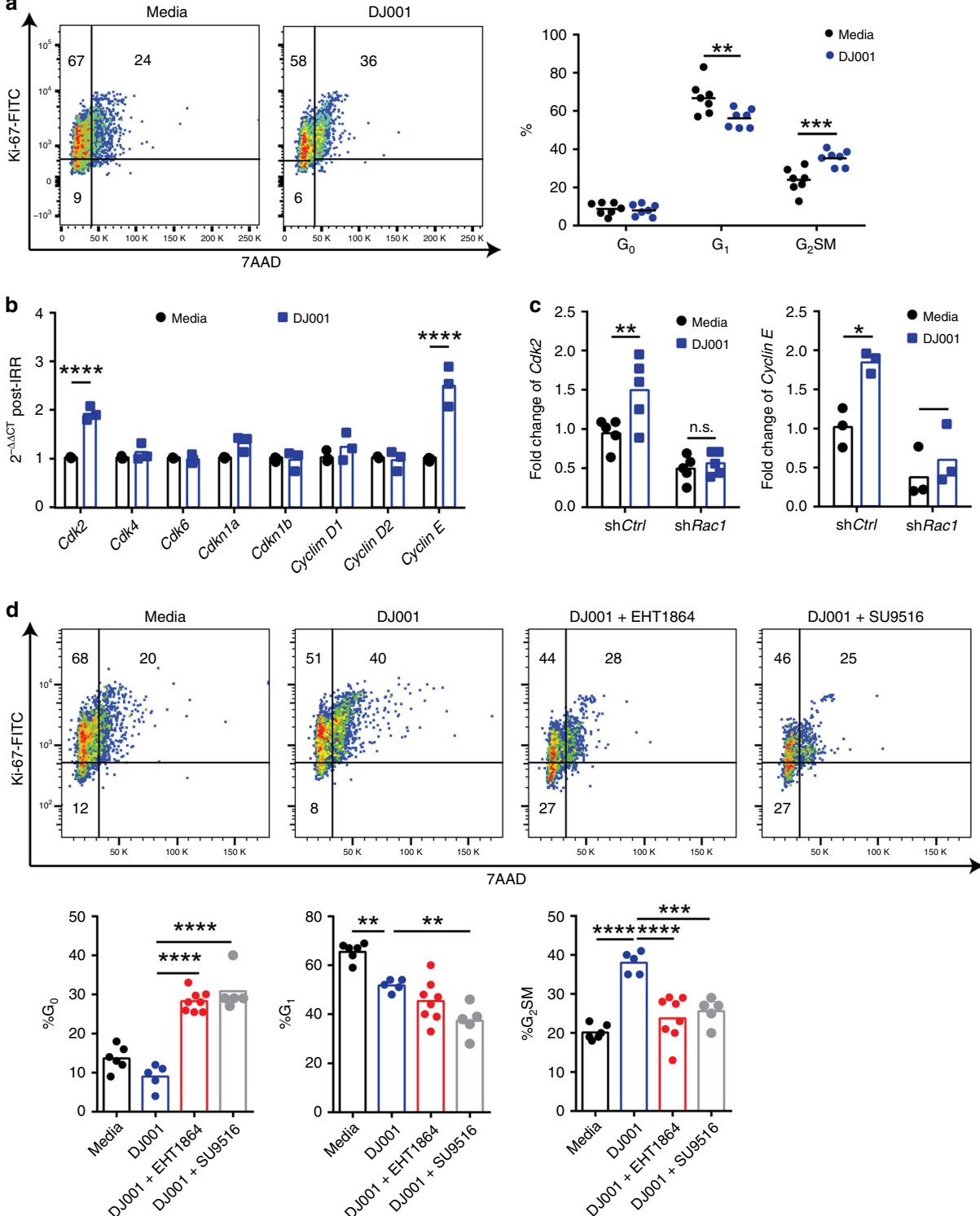

**Fig. 4** DJ001 promotes HSC proliferation via induction of CDK2. **a** At left, representative cell cycle analysis of BM KSL cells at 36 h following 300 cGy and culture with media ± 1 μg/mL DJ001. At right, mean percentages of KSL cells in $G_0$ (Ki67−7AAD−), $G_1$ (Ki67+7AAD−) and $G_2$/S/M phase (Ki67+7AAD+) are shown ($n = 7$). **b** Fold changes ($2^{-\Delta\Delta Ct}$) of cell cycle regulatory gene expression in KSL cells at 36 h following 300 cGy and culture with media ± DJ001 ($n = 3$). Gene transcript changes are normalized to *Gapdh* and media treatment. **c** Fold changes ($2^{-\Delta\Delta Ct}$) of *Cdk2* and *Cyclin E* expression in BM lin− cells in response to DJ001 at 48 h after 300 cGy, with and without *Rac1*-shRNA treatment ($n = 3$–5). **d** Top panel shows representative cell cycle analysis of KSL cells at 36 h following 300 cGy and culture with media ± 1 μg/mL DJ001, DJ001 ± 6 μg/mL EHT1864, and DJ001 ± 240 ng/mL CDK2 inhibitor SU9516. Bottom panel shows the percentages of BM cells in $G_0$, $G_1$, and $G_2$/S/M phase for each condition (media, $n = 6$; DJ001 and DJ001 + SU9516 groups, $n = 5$; DJ001 + EHT1864, $n = 8$). Two-way ANOVA with Sidak's multiple comparison test for **a**–**c**. One-way ANOVA with Tukey's multiple comparison test for **d**. Source data are provided as Source Data File. *$P < 0.05$, **$P < 0.01$, ***$P < 0.001$, ****$P < 0.0001$

following 750 cGy TBI and treatment with or without DJ001. BM KSL cells from DJ001-treated mice displayed increased bromodeoxyuridine (BrdU) incorporation compared with irradiated, vehicle-treated controls (Supplementary Fig. 7i). In order to determine whether DJ001 treatment caused durable effects on HSC cell cycle status, we measured BrdU incorporation in donor CD45.2+ cells in recipient CD45.1+ mice at day + 7 and day + 21 following competitive transplantation. Of note, donor cells were collected from the BM of CD45.2+ mice at day + 10 following 750 cGy TBI and daily treatment with DJ001 or vehicle, and then transplanted into lethally irradiated CD45.1+ recipient mice, as described in Fig. 2. We detected no difference in BrdU incorporation between donor CD45.2+ cells from either the DJ001- or vehicle-treatment groups at these time points in recipient mice (Supplementary Fig. 7j).

Deletion of Rac1 has been shown to decrease the homing, engraftment, and niche localization of hematopoietic stem and progenitor cells[30,31,36]. Therefore, it is possible that systemic treatment of mice with DJ001 altered Rac1-mediated migration and localization of endogenous HSCs in supportive BM niches, thereby facilitating hematopoietic regeneration in vivo. We observed that DJ001 treatment did not alter the in vitro migration of BM KSL cells through EC monolayers and had no effect on homing of transplanted BM ckit+lin− cells at 18 h in irradiated C57BL/6 mice (Supplementary Fig. 8a, b).

**PTPσ inhibition promotes human HSC regeneration.** As DJ001 promoted murine hematopoietic regeneration following myelosuppression, we sought to determine whether DJ001 could also accelerate human HSC regeneration. Irradiation of human BM CD34+ cells with 300 cGy in vitro caused a significant loss of CD34+CD38− cells, which are enriched for human HSCs[37], at 36 h after irradiation (Fig. 5a). Treatment with 5 µg/mL DJ001 increased the recovery of viable human BM CD34+CD38− cells following irradiation (Fig. 5a). DJ001 also increased the recovery of multipotent CFU-GEMMs at 72 h following irradiation of human BM CD34+ cells (Fig. 5b). Mechanistically, DJ001 treatment decreased radiation-induced apoptosis of human CD34+CD38− cells following irradiation (Fig. 5c). RAC inhibition blocked DJ001-mediated survival of irradiated human CD34+CD38− cells, suggesting that DJ001-mediated anti-apoptotic effects on human HSCs were RAC pathway dependent (Fig. 5c). As we observed in our murine studies, DJ001 treatment increased the expression of the pro-survival genes, *BCL2L1* and *MCL-1*, in irradiated human CD34+ cells (Fig. 5d and Table 2).

In order to determine whether PTPσ inhibition could promote the recovery of human HSCs with in vivo repopulating capacity, we irradiated human BM CD34+ cells with 300 cGy, cultured with TSF media with or without DJ001 for 36 h, and transplanted the progeny into NSG mice to assess human hematopoietic engraftment over time (Fig. 5e). Mice transplanted with irradiated, DJ001-treated BM CD34+ cells displayed substantially increased engraftment of human CD45+ cells, CD19+ B cells, CD33+ myeloid cells, and CD3+ T cells at 12 weeks post transplant compared with mice transplanted with equal doses of irradiated, control BM CD34+ cells (Fig. 5f, g). Therefore, PTPσ inhibition promoted the recovery of human HSCs with multilineage repopulating capacity following irradiation.

## Discussion

PTPs have been recognized as important potential drug targets due to their involvement in the pathogenesis of numerous diseases[38,39]. However, active site-directed PTP inhibitors frequently lack target specificity due to the high degree of homology between PTP active sites[40,41]. Negatively charged phospho-tyrosine

mimetics also have poor bioavailability, as they are typically cell impermeable[40]. These barriers have hindered the development of PTP active site inhibitors for clinical use. A recent analysis of non-receptor and receptor PTPs for "druggability" suggested that the majority of PTPs were poor drug targets due to the hydrophilic or shallow nature of their catalytic pockets[38]. As an alternative strategy, allosteric inhibitors have been described for specific PTPs, including selective inhibitors for dual specificity phosphatase 6 and the SHP2[40,42]. A small molecule, allosteric inhibitor of SHP2 was recently shown to inhibit RTK-driven human cancer growth in xenograft models[43]. Here we describe a small molecule that non-competitively inhibits PTPσ via allosteric binding and is highly selective for PTPσ. Based on these unique properties, we propose that DJ001, and DJ001 structural homologs, are compelling candidates for therapeutic development.

A large number of patients with cancer require myelosuppressive chemotherapy and/or radiotherapy in the treatment of their disease[44]. Hematopoietic toxicities occur commonly in such patients, resulting in hospitalizations, delays in curative therapy, and potentially life-threatening infectious complications[45]. Depletion of BM HSCs and progenitor cells, which occurs following repeated cycles of cytotoxic chemotherapy, contributes to prolonged myelosuppression in such patients[46]. Therapies that promote HSC and progenitor cell regeneration could lessen the complications of myelosuppression and facilitate the timely completion of potentially curative chemotherapy. However, the mechanisms that govern HSC regeneration are poorly understood and this gap in knowledge has impeded the development of targeted therapies to drive human hematopoietic regeneration. Here we demonstrate that systemic administration of a small molecule inhibitor of PTPσ accelerated hematopoietic regeneration in mice following irradiation or chemotherapy. Furthermore, PTPσ inhibition promoted the recovery of human hematopoiesis and human HSCs with long-term repopulating capacity in NSG mice following irradiation. Taken together, these results suggest the therapeutic potential for a small-molecule PTPσ inhibitor to accelerate hematologic recovery in cancer patients receiving myelosuppressive chemotherapy or radiotherapy.

Our in vitro studies of non-irradiated BM CD34−KSL cells suggested the potential for DJ001 or DJ001 analogs to promote hematopoietic progenitor cell expansion; however, in vivo administration of DJ001 3 days per week for 4 weeks caused no expansion of HSCs, progenitors, or mature blood counts in non-irradiated mice. The absence of effect in non-irradiated mice may have occurred due to insufficient concentrations of DJ001 achieved in the BM of non-irradiated mice following the chosen treatment schedule or possibly due to indirect effects of systemically administered DJ001 on cytokines or chemokines in the BM that counteracted direct effects on HSCs. Our analysis of the BM of non-irradiated mice revealed no significant changes in cytokine or chemokine levels in response to DJ001 treatment. Following TBI, DJ001 treatment was associated with modest changes in concentrations of the HSC growth factors, CXCL12 and PTN, in the BM compared with controls. These results suggest that DJ001 promotes HSC regeneration primarily via direct effects on HSCs; however, indirect effects of DJ001 via undiscovered mechanisms cannot be not excluded.

PTPσ and related receptors within the leukocyte common antigen-related (LAR) subfamily of PTPs are expressed by neurons and act as receptors for heparan sulfate proteoglycans[47–50]. In the developing nervous system, PTPσ regulates axon guidance and synapse formation[47–52]. PTPσ also serves as a receptor for inhibitory glycosylated side chains of chondroitin sulfate proteoglycans (CSPGs)[47,53–55]. CSPGs, which are enriched in the extracellular matrix surrounding injured nerves, inhibit axon regeneration following spinal cord injury[6,7]. Genetic deletion of

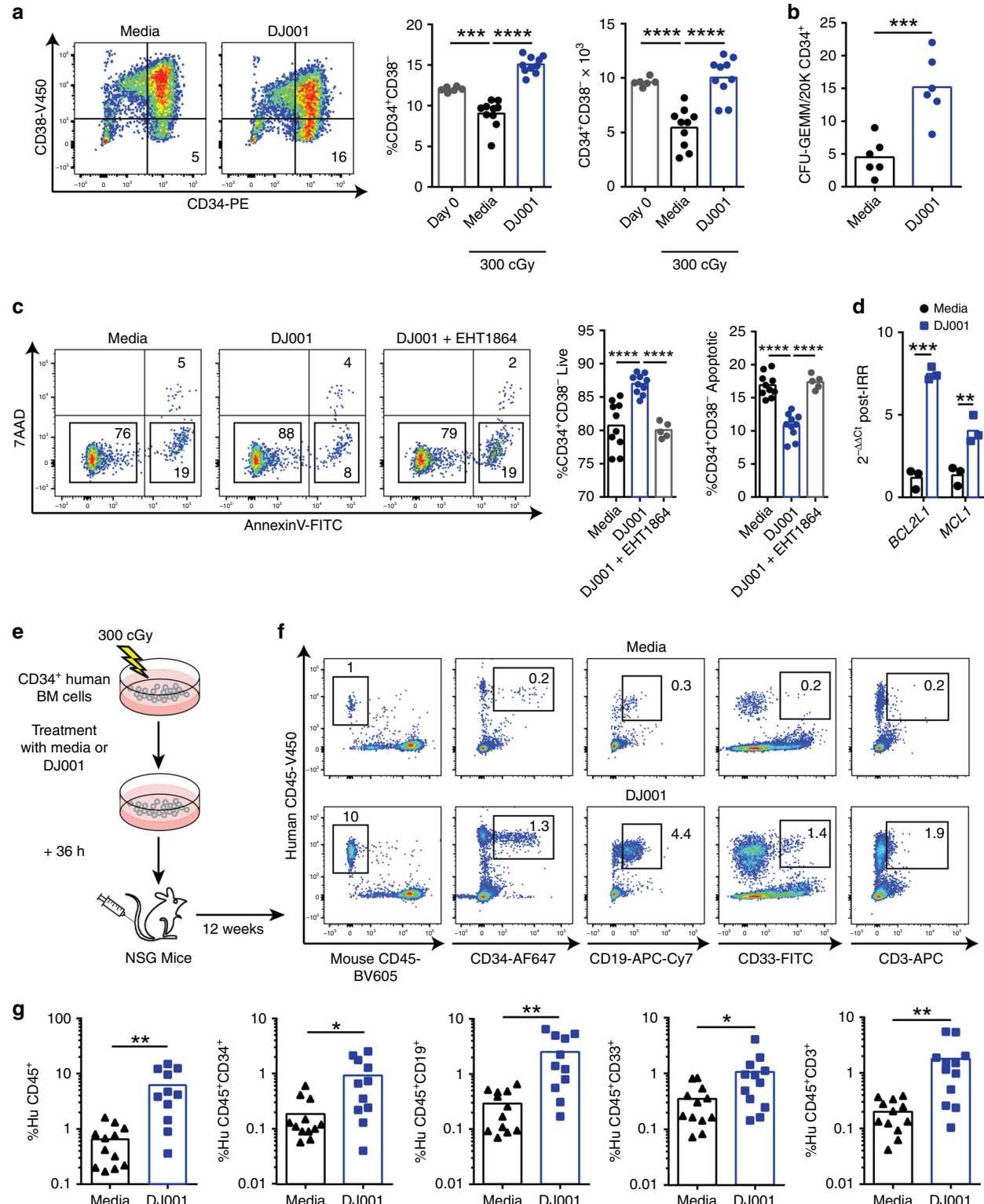

PTPσ promoted axon regeneration in a model of spinal cord injury, suggesting that PTPσ may be a therapeutic target to enhance nerve regeneration[47]. Subsequently, systemic delivery of a peptide-mimetic of the PTPσ wedge domain was reported to relieve CSPG-mediated inhibition and restore serotonergic innervation of the spinal cord in rats[53]. In a model of myocardial infarction, genetic deletion or injection of a PTPσ intracellular peptide beginning 3 days after injury promoted sympathetic innervation in the area of myocardial scar[56]. Here we

demonstrate that systemic delivery of a small-molecule PTPσ inhibitor drives HSC regeneration and hematopoietic reconstitution in vivo following myelosuppression. The combined benefits of PTPσ inhibition on hematopoietic and neural regeneration suggest an important role for PTPσ in regulating adult somatic cell regeneration.

Mechanistically, PTPσ inhibition suppressed radiation-induced apoptosis of murine and human HSCs. Downregulation of LAR protein, which, along with PTPσ and PTPδ, comprises the Type

**Fig. 5** DJ001 promotes human HSC regeneration following irradiation. **a** At left, representative flow cytometric analysis of CD34$^+$CD38$^-$ cells at 36 h after 300 cGy irradiation of BM CD34$^+$ cells and culture with media ± 1 μg/mL DJ001. At right, mean percentages and numbers of CD34$^+$CD38$^-$ cells in each group (Day 0, $n = 6$; media, DJ001, $n = 10$). One-way ANOVA with Tukey's multiple comparison test. **b** Numbers of CFU-GEMMs within cultures of CD34$^+$ cells at 72 h after 300 cGy and culture with media ± DJ001 ($n = 6$). **c** At left, representative flow cytometric analysis of Annexin V/7AAD staining of CD34$^+$ cells at 24 h after 300 cGy and culture with media ± DJ001 ± EHT1864. At right, percentages of live and apoptotic CD34$^+$CD38$^-$ cells in each condition (media and DJ001, $n = 10$; DJ001 + EHT1864, $n = 5$). One-way ANOVA with Tukey's multiple comparison test. **d** Fold changes ($2^{-\Delta\Delta Ct}$) in *BCL2L1* and *MCL-1* gene expression in CD34$^+$CD38$^-$ cells at 12 h after 300 cGy in media ± DJ001 ($n = 3$). Gene transcript changes were normalized to *GAPDH* and media treatment. Two-way ANOVA with Sidak's multiple comparison test. **e** Schematic representation of NSG mice transplantation assay using the progeny of human BM CD34$^+$ cells irradiated with 300 cGy and treated with or without DJ001 × 36 h. **f** Representative flow cytometric analysis of human CD45$^+$ cells, human CD34$^+$ cells, human CD19$^+$ B cells, human CD33$^+$ myeloid cells, and human CD3$^+$ T cells engraftment in the BM of NSG mice at 12 weeks post transplantation. **g** Percent engraftment of human hematopoietic cell subsets in the BM of NSG mice at 12 weeks post transplantation ($n = 11$–$12$/group). Source data are provided as Source Data File. *$P < 0.05$, **$P < 0.01$, ***$P < 0.001$, ****$P < 0.0001$

---

**Table 2 Primers used for human gene detection**

| Target mRNA | Sequence 5′–3′ |
| --- | --- |
| hGAPDH Fw | CCTGCACCACCAACTGCTTA |
| hGAPDH Rv | GGCCATCCACAGTCTTCTGAG |
| hBCL2L1 Fw | GACTGAATCGGAGATGGAGACC |
| hBCL2L1 Rv | GCAGTTCAAACTCGTCGCCT |
| hMCL-1 Fw | TGCTTCGGAAACTGGACATCA |
| hMCL-1 Rv | TAGCCACAAAGGCACCAAAAG |

---

IIa receptor PTPs, has similarly been shown to have pro-survival effects on neurons in the setting of serum deprivation[57,58]. Our results suggest that PTPσ inhibition promotes HSC survival following irradiation via RAC pathway activation and induction of BCL-X$_L$ in HSCs. Treatment with the PTPσ inhibitor, DJ001, uncoupled PTPσ binding to p250GAP, causing RAC1 activation in primary BM HSCs, without affecting other RhoGTPases, CDC42 and RhoA. Deletion of *Rac1* in the hematopoietic compartment of mice has been shown to cause decreased HSPC engraftment, homing, and niche localization[30,31,36], whereas expression of a dominant-negative *Rac2* increased HSPC apoptosis[36,59]. RAC1 has also been shown to inhibit apoptosis of irradiated breast cancer cells, epithelial cells, and T cells via induction of BCL-X$_L$, MCL-1, and other BCL-2-like proteins[32,60–62]. Here, PTPσ inhibition with DJ001 increased the expression and protein levels of BCL-X$_L$ in HSCs and suppressed HSC apoptosis following irradiation, dependent upon RAC pathway and ERK1/2 activation. These results suggest that RAC pathway activation and induction of BCL-X$_L$ regulate HSC survival in response to PTPσ inhibition.

In addition to regulation of HSC survival, our data suggest that DJ001 promoted early HSPC proliferation compared with irradiated, control HSCs. Further, DJ001-mediated HSC cell cycle progression was dependent on RAC pathway activation and induction of CDK2. These findings are consistent with the observations by Gu et al.[30], who showed that *Rac1*$^{-/-}$ HSPCs displayed decreased entry into G$_2$/S/M phase in response to SCF. As PTPσ regulates the phosphorylation of several kinases and the RAC pathway regulates multiple pathways, PTPσ inhibition may have altered the expression of other effectors that regulate HSC proliferation beyond CDK2. However, we did not observe any durable effect of DJ001 treatment on the proliferative potential of HSCs or their progeny based on BrdU incorporation analysis of engrafted donor CD45.2$^+$ cells over time following transplantation into recipient CD45.1$^+$ mice. It is therefore perhaps more likely that the anti-apoptotic effects of systemic DJ001 treatment on HSCs in irradiated mice contributed to the increase in recovery of phenotypic HSPCs as well as HSCs with in vivo repopulating capacity. It is also possible that DJ001 treatment

favorably affected HSC transcriptional programs that regulate myeloid and lymphoid cell differentiation and HSC repopulating potential[63].

The paucity of therapeutics capable of accelerating HSC regeneration and hematopoietic reconstitution in myelosuppressed patients highlights an unmet medical need. Here we describe a class of selective, allosteric PTPσ inhibitors that promote the regeneration of murine and human HSCs capable of long-term hematopoietic reconstitution. Our results provide the mechanistic foundation for the development of selective PTPσ inhibitors to promote hematopoietic regeneration in patients receiving myelosuppressive chemotherapy, radiotherapy, and those undergoing myeloablative hematopoietic cell transplantation.

## Methods

**Animal models**. All animal procedures were performed in accordance with animal use protocols approved by the UCLA Animal Research Committee (Approval Number 2014-021-13 M). *Ptprs*$^{-/-}$ mice were provided by Dr Michel Tremblay (McGill University). C57BL/6 mice, B6.SJL mice, and NOD.Cg-*Prkdc*$^{scid}$*Il2rg*$^{tm1Wjl}$/SzJ (NSG) mice between 8 and 12 weeks old were obtained from the Jackson Laboratory.

**Synthesis of DJ001**. A mixture of 1-phenylprop-2-yn-1-one* (0.303 g, 2.3 mmol), 3-nitroaniline (0.427 g, 3.0 mmol), and copper (I) iodide (0.078 g, 0.4 mmol) in dimethylformamide (6 mL) and water (60 μL) was stirred at 85 °C for 16 h. After 16 h, the reaction mixture was allowed to cool down to 21 °C and was diluted with water (50 mL), and extracted with ethyl acetate (3 × 50 mL). The combined organic layer was washed with NH$_4$Cl/NH$_3$ (1:1 v/v, 3 × 20 mL) and brine (30 mL), dried over MgSO$_4$, filtered, and concentrated under reduced pressure. The crude residue was purified by column chromatography (*n*-hexane/ethyl acetate = 20:1) to obtain (Z)-3-((3-nitrophenyl)amino)-1-phenylprop-2-en-1-one (DJ001, 140 mg, 0.52 mmol, 23%) as a yellow powder. $^1$H NMR (400 MHz, CDCl$_3$) δ 12.26 (d, NH, $J = 11.6$ Hz), 7.96–7.91 (m, 3 H), 7.90 (ddd, 1 H, $J = 8.0, 2.0, 0.8$ Hz), 7.56–7.46 (m, 5 H), 7.37 (ddd, 1 H, $J = 8.0, 2.4, 0.8$ Hz), 6.16 (d, 1 H, $J = 8.0$ Hz); $^{13}$C NMR (100 MHz, CDCl$_3$) δ 191.8, 149.4, 143.4, 141.6, 138.6, 132.2, 130.6, 128.6, 127.5, 122.3, 117.8, 110.0, 95.7. High resolution mass spectrometry (HRMS) m/z calculated for C$_{15}$H$_{13}$N$_2$O$_3$ [M + H] + 269.0904, found 269.0921. *1-Phenylprop-2-yn-1-one was prepared by oxidation of 1-phenylprop-2-yn-1-ol in a two-step procedure from benzaldehyde and trimethylsilylacetylene by a known method[64].

**Synthesis of DJ009**. (Z)-3-((3,5-difluorophenyl)amino)-1-phenylprop-2-en-1-one (DJ009) was prepared by the same procedure using 3,5-difluoroaniline. $^1$H NMR (400 MHz, CDCl$_3$) δ (p.p.m.): 12.0 (d, $J = 11.7$ Hz, 1 H), 7.94 (dt, $J = 6.9, 1.6$ Hz, 2 H), 7.55–7.50 (m, 2 H), 7.48–7.45 (m, 2 H), 7.38 (dd, $J = 11.7, 8.2$ Hz, 1 H), 6.64–6.57 (m, 2 H), 6.54–6.47 (m, 1 H), 6.09 (d, $J = 8.2$ Hz, 1 H); $^{13}$C NMR (100 MHz, CDCl$_3$) δ (p.p.m.): 191.7, 164.0 (dd, $J = 249, 15$ Hz), 143.5, 142.8 (t, $J = 13$ Hz), 138.7, 132.1, 128.6, 127.5, 99.3 (dd, $J = 20, 8$ Hz), 98.5 (t, $J = 26$ Hz), 95.3; $^{19}$F NMR (CDCl$_3$, 376 MHz) δ (p.p.m.): −107.9. HRMS m/z calcd. for C$_{15}$H$_{12}$F$_2$NO [M + H] + 260.0881, found 260.0866.

**Phosphatase assay**. The two intracellular catalytic domains (D1 and D2) of *Ptprs* were cloned into a pET28a vector, overexpressed in *Escherichia coli* BL21 and purified[65]. Enzymatic activity of PTPσ was assayed using a modified version of the Malachite Green Assay[66] and the Tyrosine Phosphatase Assay Kit (Promega Corporation). Unless stated otherwise, standard assays were carried out using 50 nM PTPσ protein in 1× Buffer (10 mM Tris, 5 mM MgCl$_2$, 10 mM NaCl, 0.02% Tween) and Tyr Phosphopeptide as substrate (200 μM for Fig. 1b, c; 50–1200 μM in Fig. 1c). Purified catalytic domain 1 and domain 2 of PTPσ were pre-incubated

with test compounds 3071, 5205, 5075, or control for 15 min in a 96-well plate before the addition of Tyr Phosphopeptide (DADE(pY)LIPQQG). For IC$_{50}$ determination, rates normalized relative to uninhibited controls (dimethyl sulfoxide (DMSO)) were plotted against compound concentration and fitted using a four-parameter nonlinear regression curve fit ($y = [(A - D) (1 + \{xC - 1\}B) - 1] + D)$ (Prism 6.0, Graphpad Software). For mechanism studies and determination of the enzyme's $K_m$ and $V_{max}$, data were analyzed using a nonlinear regression fit according to the classical Michaelis–Menten kinetics model, $Y = V_{max}*X/(K_m + X)$ (Prism 6.0, Graphpad Software).

**Phosphatase Profiler screen**. Compound DJ001 was evaluated in a PhosphaseProfiler™ screen at 10 μM and 1 μM (2.7 μg/mL and 0.27 μg/mL) concentrations at Eurofins Pharma Discovery Services UK (Study number UK022-0004033) against a panel of 21 Phosphatases. In each experiment, the respective reference antagonist/agonist was tested directly with DJ001 and the data were compared with historical values determined at Eurofins. DJ001 compound inhibition was calculated as percentage inhibition of the enzymatic activity compared with control.

**Pharmacokinetic study for DJ001**. PK studies for DJ001 were performed at Cyprotex (Study Number CYP1426-R1) under non-good laboratory practice conditions. Mice were subcutaneously injected with a single dose of 5 mg/kg DJ001 (in 10% DMSO, 40% phosphate-buffered saline (PBS), 50% polyethylene glycol and plasma samples were collected at 0.08, 0.25, 0.5, 1, 2, 4, 8, and 24 h. Samples were crashed with three volumes of methanol and analytical internal standard (propranolol). Supernatant was subjected to liquid chromatography-mass spectrometry/mass spectrometry (MS/MS) using an acetonitrile-water gradient system and electrospray ionization in multiple reaction monitoring mode for MS detection. All plasma samples were compared with an internal calibration curve prepared in mouse blank plasma.

**Proximity ligation assay**. A PLA assay was performed by using the Duolink In Situ Red Starter Kit (Millipore Sigma) to detect the binding between PTPσ and p250GAP proteins. BM KSL cells from C57BL/6 mice were sorted by fluorescence-activated cell sorting and then plated onto fibronectin pre-coated chamber slides overnight. The next day, cells were treated with 1 μg/mL DJ001 or vehicle (equal volume of DMSO) for 1 h at 37 °C. Cells were then fixed with 4% paraformaldehyde (PFA) for 20 min. The slides were blocked with Duolink blocking solution and incubated with rabbit anti-ARHGAP32 polyclonal antibody (Bioss Antibodies, #bs-9296R, 1:50 dilution) and goat anti-PTPσ (K-19) polyclonal antibody (Santa Cruz Biotechnology, #sc-10873, 1:50) overnight at 4 °C. Cells were imaged using a Leica SP8 Confocal Microscope equipped with a ×63 objective lens. A white light laser set to 580 nm was used to excite dsRed and a UV laser was used to excite 4′,6-diamidino-2-phenylindole (DAPI). Analysis was performed using the spots detection function in IMARIS 9.0.2 (Bitplane).

**p250GAP phospho-tyrosine sandwich ELISA**. Clear, pre-blocked Protein A-coated Microtiter wells (Fisher Scientific, #15132) were incubated with ARHGAP32 polyclonal antibody (Bioss, #bs-9296R, 1:500) in antibody dilution buffer (150 mM NaCl, 25 mM HEPES pH 7.2, 0.5% bovine serum albumin, 0.05% Tween 20) for 3 h at room temperature (RT). Subsequently, the plate was washed three times and phosphorylated p250GAP from BM lin⁻ cell lysate was added to each well. BM lin⁻ cells were collected in NP40 buffer (ThermoFisher Scientific; FNN0021) supplemented with 1× complete, Mini, EDTA-free Protease Inhibitor Cocktail (Sigma Aldrich; 4693159001) and 1× PhosSTOP (Sigma Aldrich; 4906845001). Following incubation for 2 h at 4 °C, the plate was completely emptied (without washing) and exposed to 50 μL of a fixation solution (0.5% formaldehyde in 300 mM NaCl, 20 mM sodium phosphate buffer, pH 7) for 10 min. The plate-bound, phosphorylated p250GAP was measured by a specific anti-phospho-tyrosine peroxidase-coupled antibody (Sigma Aldrich, #A5964-1VL, 1:500) in antibody dilution buffer. Incubation was for 1 h at RT. Peroxidase activity was measured by a Tecan Infinity plate reader using BM Chemiluminescence ELISA Substrate (Sigma Aldrich, #11582950001).

**G-LISA activation assays**. The RAC1-GTP, RHOA-GTP, and CDC42-GTP activation levels in BM lin⁻ cells were measured using a colorimetric-based RAC1-, RHOA-, and CDC42 G-LISA Activation Assay Kit (Cytoskeleton, Inc.). BM cells from femurs and tibias were isolated from 12-week-old $Ptprs^{+/+}$ and $Ptprs^{-/-}$ mice. Cells were then depleted of lineage-committed cells with Direct Lineage Cell Depletion Kit (Miltenyi Biotec, #130-110-470; 1:5). The BM lin⁻ cell fraction was then serum starved in Iscove's modified Dulbecco's medium (IMDM) and treated with either vehicle (equal amount of DMSO) or 1 μg/mL DJ001 for 10 min at 37 °C. After treatment, cells were washed with ice-cold PBS and then placed in lysis buffer supplemented with protease inhibitor. Lysate concentrations were measured by Pierce™ BCA Protein Assay Kit (ThermoFisher Scientific). G-LISA was performed according to the manufacturer's instructions. Briefly, 12.5 μg of lysates was added to a GTP-binding protein pre-coated plate and active RAC1-GTP, RHOA-GTP, or CDC42-GTP levels were measured at 490 nm using a PowerWave XS2 microplate reader (BioTek).

**Lentivirus-mediated shRNA silencing**. $Rac1$ CDS-targeting- and $Bcl2l1$ CDS-targeting shRNA in lentiviral plasmid (TRCN0000310888 and TRCN0000004685) and control shRNA (SHC216V) were purchased from Sigma Aldrich. For viral production, 293T cells were transfected with lentiviral gag/pol and VSV-G (courtesy of Donald Kohn, UCLA) and the lentiviral plasmids, at a ratio of 2.2: 1.2: 3.3 (in [μg], gag/pol:VSVG:Plasmid) using Lipofectamine 3000 and P300 Enhancer. Viral particles were collected after 24 h and 48 h. One milliliter of viral supernatant was used to infect $0.75 \times 10^6$ BM lin⁻ cells. Infected cells were collected the next day and irradiated with 300 cGy followed by treatment with or without 1 μg/mL DJ001, prior to CFC assay.

**Immunofluorescence microscopy**. Lab-Tek chamber slides were coated with fibronectin (25 μg/mL, Millipore Sigma #341635). Sorted BM KSL cells ($1 \times 10^4$) were resuspended in 200 μL TSF media and added to each of the pre-coated wells and incubated overnight (37 °C/5% CO$_2$). The next day, cells were serum starved for 30 min and then stimulated with DJ001 (1 μg/mL), EHT1864 (6 μg/mL), or equal volumes of DMSO for 5 min. Cells were washed once with PBS and fixed with 4% PFA for 10 min. Cells were permeabilized with 0.5% Triton/PBS (PRM) for 30 min and then blocked with 5% fetal bovine serum (FBS)/PRM for 1 h. The slide was then incubated with a phospho-ERK1/2 primary antibody (Cell Signaling, #4377, 1:200) for 1 h. The wells were washed three times and then incubated with a donkey anti-rabbit Alexa Fluor 488 secondary antibody (Thermo Fischer, #A21206, 1:200) at a 1:200 concentration for 1 h. The slide was washed three times and mounted with ProLong Gold Antifade Reagent with DAPI. Cells were imaged using a ×63 objective on a Zeiss Axio Imager M2 widefield fluorescence microscope with all microscope settings derived from imaging a secondary only control. Data were analyzed using Fiji (ImageJ). Briefly, the cell outline was identified by threshold levels and the mean fluorescence intensity within the cell area was quantified.

**Flow cytometric analysis**. Femurs and tibiae were collected from killed C57BL/6 or $Ptprs^{-/-}$ mice and flushed with IMDM containing 10% FBS and 1% penicillin–streptomycin for BM cells. PB was collected through sub-mandibular puncture. Cells were filtered through a 40 μM strainer and then treated with ACK lysis buffer (Sigma Aldrich) before antibody staining for flow cytometry. For KSL and CD150⁺CD48⁻KSL cell analysis, BM cells were stained with allophycocyanin (APC)- and Cy7-conjugated anti-mouse Sca1 (BD Biosciences, #560654, 1:100), phycoerythrin (PE)-conjugated anti-mouse c-kit (BD Biosciences, #553355, 1:100), V450 lineage cocktail (BD Biosciences, 561301, 1:10), Alexa Fluor 488-conjugated anti-mouse CD48 (BioLegend, #103414, 1:100), and Alexa Fluor 647-conjugated anti-mouse CD150 (Biolegend, #115918, 1:100) antibodies. For MEP, CMP, GMP, and CLP cell analysis, BM cells were stained with APC-Cy7-conjugated anti-mouse Sca1 (BD Biosciences, #560654, 1:100), PE-conjugated anti-mouse ckit (BD Biosciences, #553355, 1:100), V450 lineage cocktail (BD Biosciences, #561301; 1:10), Alexa Fluor 488-conjugated anti-mouse CD127 (BD Biosciences, #561533, 1:100), Alexa Fluor 647-conjugated anti-mouse CD34 (BD Biosciences, #560230; 1:100), and BV605-conjugated anti-mouse CD16/32 antibodies (BD Biosciences, #563006; 1:100). For donor engraftment analysis in transplanted mice, PB or BM cells were stained with BV605-conjugated anti-mouse CD45.2 (BioLegend, #109841, 1:100), fluorescein iso-thiocyanate (FITC)-conjugated anti-mouse CD45.1 (BD Biosciences, #553775, 1:100), PE-conjugated anti-mouse Mac-1 (BD Biosciences, #557397, 1:100) and anti-mouse Gr1 (BD Biosciences, #553128, 1:100), V450-conjugated anti-mouse CD3 (BD Biosciences, #561389, 1:100), and APC-Cy7-conjugated anti-mouse B220 (BD Biosciences, 552094, 1:100) antibodies.

Intracellular flow cytometric analysis was performed on irradiated (300 cGy) or non-irradiated, sorted KSL cells after treatment with 1 μg/mL DJ001 or control (equal volumes of DMSO) for 24 h. At 24 h after irradiation, cells were fixed with 4% PFA for 10 min, followed by permeabilization using 0.25% saponin in PBS. Cells were washed again and stained with antibody at the recommended concentrations for 30 min at RT. Intracellular antibodies and phospho-flow antibodies used were as follows: FITC-conjugated anti-BCL-X$_L$ (Abcam, #26148, 1:100), active RAC1-GTP antibody (NewEast Biosciences, #26903, 1:100) and anti-PAK1 (phospho S144) + PAK2 (phospho S141) + PAK3 (phospho S154) antibody (Abcam, #40795, 1:100), and FITC-conjugated goat anti-rabbit IgG H&L (Abcam, #97050, 1:200).

**Gene expression analysis**. For all studies, RNA was isolated using the Qiagen RNeasy micro kit (Qiagen). RNA was reverse transcribed using the High-Capacity cDNA Reverse Transcription Kit (ThermoFisher Scientific) and was then used for quantitative PCR with SYBR Select Master Mix (Life Technologies). Values were normalized to housekeeping gene $Gapdh/GAPDH$ and given as ΔΔCt values normalized to media-treated, non-irradiated BM KSL cells ($2(-\Delta\Delta C(T))$ method)[67].

**Survival studies**. For survival studies, 10-week-old female C57BL/6 mice were irradiated with 750 cGy TBI, which is lethal for ~50% of C57BL/6 mice by day + 30 (LD50/30), using a Shepherd Cesium-137 irradiator. Twenty four hours post irradiation, mice were administered daily subcutaneous injections of 5 mg/kg DJ001 or DJ009, or vehicle in a volume of 100 μL for 10 days. DJ001 or DJ009 injections were prepared in PBS, 0.5% Tween 80, and 10% DMSO. Corresponding

vehicle injections contained PBS, 10% DMSO, and 0.5% Tween 80. PB complete blood counts were measured using a Hemavet 950 instrument (Drew Scientific) at day + 10 post irradiation. For hematopoietic analysis, BM cells were collected at day + 10 post irradiation. To study whether DJ001 increased survival rates through activation of RAC signaling, the RAC inhibitor, EHT1864 (Selleckchem), was dissolved in PBS and administered intraperitoneally, 40 mg/kg every other day, to 750 cGy-irradiated mice until day + 10.

**Chemotherapy model.** Ten-week-old female C57BL/6 mice received a single tail vein injection of 250 mg/kg 5FU in 100 μL PBS. Two hours post 5FU injection, mice were administered daily subcutaneous injections of 5 mg/kg (100 μg per mouse) DJ001 or vehicle in a volume of 100 μL for 8 days. DJ001 injections were prepared in PBS, 50% PEG-400, and 10% DMSO. Corresponding vehicle injections contained PBS, 50% PEG-400, and 10% DMSO. Retro-orbital blood was collected on day + 8, + 11, and + 14 after 5FU treatment with capillary pipettes (Fisher Scientific). Complete blood counts were measured using a Hemavet 950 instrument (Drew Scientific).

**Cytokine analysis.** For in vitro cytokine measurements, $3.5 \times 10^4$ BM KSL cells were irradiated with 300 cGy and cultured in TSF media with and without 1 μg/mL DJ001. Non-irradiated KSL cells were cultured in TSF media with and without 1 μg/mL DJ001. At 12 h of culture, culture supernatants were collected for analysis. For in vivo cytokine measurements, adult C57BL/6 mice were irradiated with 750 cGy TBI and then treated with 5 mg/kg of DJ001 or vehicle, subcutaneously, from day + 1 to day + 5. Non-irradiated mice received the identical regimen of 5 mg/kg DJ001 or vehicle for 5 days. Mice were killed and BM cells were collected into IMDM. Cells were pelleted and the BM supernatant was collected and sent to the UCLA Immune Assessment Core for cytokine analysis. All samples were tested on a mouse 32-plex Cytokine/Chemokine panel platform from EMD Milipore using Luminex's xMAP® immunoassay technology (Luminex, Inc.). In addition, the following cytokine levels were assessed by solid-phase ELISA assay according to the manufacturer's description: CXCL12 (R&D Systems), SCF (R&D Systems), Thrombopoietin (TPO) (R&D Systems), PTN (Biomatik), DKK1 (R&D Systems), and EGF (R&D Systems).

**In vivo BrdU incorporation analysis.** For BrdU incorporation analysis in BM KSL cells in vivo, adult C57Bl/6 mice were irradiated with 750 cGy TBI and treated with 5 mg/kg DJ001 or vehicle daily from day + 1 to day + 10. On day + 10, mice were injected intraperitoneally with 2 mg BrdU (BD Biosciences). Sixteen hours later, BM cells were collected and stained with anti-BrdU FITC (BD Biosciences, #557891, 1:50), APC-Cy7-conjugated anti-Sca1 (BD Biosciences, #560654, 1:100), PE-conjugated anti-c-kit (BD Biosciences, #553355, 1:100), and V450 lineage cocktail (BD Biosciences, #561301, 1:10). We repeated this BrdU incorporation analysis on donor CD45.2+ cells at day + 7 and day + 21 following competitive transplantation into recipient CD45.1+ mice.

**Apoptosis assay and cell cycle analysis.** Eight- to ten-week-old female C57BL/6 mice were irradiated with 500 cGy TBI followed by subcutaneous injections of 5 mg/kg DJ001 or vehicle, or 5 mg/kg DJ001 plus 40 mg/kg EHT1864. Twenty four hours after irradiation, mice were killed to collect BM cells for Annexin V apoptosis analysis. BM cells were stained with APC-Cy7-conjugated anti-mouse Sca1 (BD Biosciences, #560654, 1:100), PE-conjugated anti-ckit (BD Biosciences, #553355, 1:100), and V450 lineage cocktail (BD Biosciences, #561301, 1:10) antibodies for 30 min at 4 °C. Cells were then resuspended in 1× Annexin V binding buffer and then stained with FITC-conjugated anti-Annexin V antibody (BD Biosciences, #556547,1:50) and 7AAD antibody (BD Biosciences, #559925, 1:40)[68]. For in vitro human BM cell apoptosis assays, isolated human BM CD34+ cells were plated into 96-well plates and then irradiated with 300 cGy. Immediately after irradiation, cells were treated with 5 μg/mL DJ001 or vehicle for 24 h and then collected for Annexin V cell apoptosis analysis. For cell cycle analysis, BM KSL cells were irradiated at 300 cGy and then treated with 1 μg/mL DJ001 or vehicle for 36 h. Subsequently, cells were resuspended in 200 μL of PBS, fixed with 1 mL of ice-cold fixation solution (0.25% Saponin, 2.5% PFA, 2% fetal calf serum (FCS) in PBS), and incubated for 30 min on ice. Cells were washed in Saponin buffer (0.25% Saponin, 2% FCS in PBS) and resuspended in 200 μL of Saponin buffer containing 20 μL of FITC-conjugated anti-Ki67 antibody (BD Biosciences, #556026, 1:10) and 1 μL of RNase (Qiagen, #19101, 100 mg/mL). After 30 min of incubation, cells were washed in Saponin buffer and resuspended for flow analysis in 200 μL of Saponin buffer containing 5 μL 7AAD (BD Biosciences, #559925, 1:40).

**Isolation of BM HSCs.** BM cells were first treated with ACK lysis buffer (Sigma Aldrich) and lineage-committed cells were removed using a Direct Lineage Cell Depletion Kit (Miltenyi Biotec, #130-110-470, 1:5). Lin− cells were stained with APC-Cy7-conjugated anti-mouse Sca1 (BD Biosciences, #560654, 1:100), PE-conjugated anti-ckit (BD Biosciences, #553355, 1:100), V450 lineage cocktail (BD Biosciences, #561301, 1:10), and FITC-conjugated anti-CD34 (BD

Biosciences, #553733, 1:100) antibodies. Sterile cell sorting was conducted on a BD FACS-Aria cytometer. Purified KSL cells and CD34−c-kit+sca-1+Lin− (CD34−KSL) cells were collected into IMDM (Life Technologies) + 10% FBS + 1% penicillin–streptomycin.

**CFC assays and HSC competitive repopulation assays.** CFC assays (colony-forming unit-granulocyte monocyte), burst-forming unit-erythroid, and CFU-GEMM were performed using MethoCult GF M3434 (Stemcell Technologies)[6]. For all in vitro assays, BM CD34−KSL cells, KSL cells, and Lin− cells were cultured in TSF media (IMDM, 10% FBS, 1% pen-strep, 20 ng/mL recombinant mouse TPO, 125 ng/mL recombinant mouse SCF, 50 ng/mL recombinant mouse Flt3 ligand). Recombinant mouse SCF, Flt3 ligand, and TPO were purchased from R&D Systems. For competitive repopulation assays, BM cells were isolated from donor 10–12-week-old female CD45.2+ mice. Recipient 10-week-old female CD45.1+ B6. SJL mice were irradiated with 950 cGy TBI using a Cs137 irradiator and donor BM cells were administered via tail vein injection along with a competing dose of $1 \times 10^5$ non-irradiated host BM cells. Multilineage donor hematopoietic cell engraftment was measured in the PB by flow cytometry[69].

**Transendothelial migration assay.** For the transendothelial migration assay, VeraVec™ mouse spleen ECs (Angiocrine Biosciences) were cultured to confluence in 8 μM pore transwells (Corning Incorporated). Transwells were then seeded with 50,000 sorted BM KSL cells in IMDM with 10% FBS, 1% pen-strep, 20 ng/mL TPO, 125 ng/mL SCF, and 50 ng/mL Flt3 ligand with or without 1 μg/mL of DJ001. SDF-1, 500 ng/mL (R&D Systems), was added to the bottom chamber of the transwell. At 18 h post incubation, cells in the bottom chamber were collected and cell counts were performed.

**Homing study.** Sorted BM sca-1+lin− cells (100,000) from DsRed+ mice were transplanted into lethally irradiated (950 cGy) recipient C57BL/6 mice followed by subcutaneous treatment of 5 mg/kg DJ001 or vehicle × 1 dose. At 18 h post transplantation, BM cells were collected from the recipient mice and were analyzed for percentages of DsRed+ cells in the BM[17].

**Human BM cultures and human BM transplantation assays.** Human BM mononuclear cells were purchased from AllCells. Cryopreserved human BM cells were recovered in IMDM + 10% FBS + 1% penicillin–streptomycin and then positively selected for CD34+ stem/progenitor cells by using CD34 MicroBead Kit (Miltenyi Biotec, #130-046-702, 1:5). CD34+ cells were cultured in human TSF media (IMDM, 10%FBS, 1% pen-strep, 20 ng/mL recombinant human TPO, 125 ng/mL recombinant human SCF, 50 ng/mL recombinant human Flt3 ligand (R&D Systems). The progeny of $2 \times 10^5$ irradiated, human BM CD34+ cultured for 36 h and treated with DJ001 at 5 μg/mL, were transplanted via tail vein injection into 10–12-week-old NSG mice preconditioned with 275 cGy TBI. Multilineage donor hematopoietic cell engraftment was monitored in the PB and BM by flow cytometry[20]. The PB or BM cells were stained with BV605-conjugated anti-mouse CD45 (BioLegend, #103139, 1:100), AF647-conjugated anti-human CD34 (BioLegend, #343618, 1:100), FITC-conjugated anti-human CD33 (BD Biosciences, #555626, 1:100), V450-conjugated anti-human CD45 (BD Biosciences, #560368, 1:100), APC-conjugated anti-human CD3 (BD Biosciences, #555342, 1:100), and APC-Cy7-conjugated anti-human CD19 (BD Biosciences, #557791, 1:100) antibodies.

**In silico molecular docking studies.** Molecular docking of DJ001, (Z)-isomer, to PTPσ (PDB ID: 2FH7) was carried out by AutoDock Vina, in which the Iterated Local Search Globule Optimizer was applied as optimization algorithm[70]. Each structure of ligand was prepared in Maestro 10.5 (Schroedinger, LLC) and minimized with the OPLS_2005 force field. All hydrogen atoms were added to each protein and ligand to be docked and each coordinate file of protein and ligand was generated as PDBQT file using AutoDockTools-1.5.6. A grid box for binding site was set as 18 Å in the three dimensions (x, y, and z) that covered the catalytic site of the protein or 40 Å in the three dimensions for allosteric binding site. The box had 1.0 Å grid spacing and centered at the geometric center of the protein. In each docking experiment, the best binding mode was selected according to the binding affinity calculated by the scoring function in AutoDock Vina. Docking results were analyzed with PyMOL (Schroedinger LLC, NY) and visualized by VMD 1.9.2 (UIUC, IL).

**Statistical analysis.** GraphPad Prism 6.0 was used for all statistical analyses. All data were checked for normal distribution and similar variance between groups. Data were derived from multiple independent experiments from distinct mice or culture plates. Sample sizes for in vitro studies were chosen based on observed effect sizes and SEs from prior studies. For all animal studies, a power test was used to determine the sample size needed to observe a twofold difference in means between groups with 0.8 power. A two-tailed Student's t-test was utilized for all comparison, except where otherwise noted in the figure legends. All animal studies were performed using sex- and age-matched animals, with wild-type littermates as controls. Animal studies were performed without blinding of the investigator.

Values are reported as means ± SEM, unless stated otherwise. Results were considered significant when $P < 0.05$.

**Reporting summary**. Further information on research design is available in the Nature Research Reporting Summary linked to this article.

## Data availability

The authors declare that all data supporting the findings of this study are available within the article and its Supplementary Information files or from the corresponding author upon reasonable request. The source data underlying Figs. 1b, c, e, 2, 3, 4, and 5 are provided as a Source Data File.

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

## Acknowledgements

We thank M. Tremblay (McGill University, Montreal, CA) for the *Ptprs*$^{-/-}$ mice. This work was supported, in part, by NIAID grant AI-067769-12 (J.P.C.), NHLBI grant HL-086998-08 (J.P.C.), the California Institute for Regenerative Medicine Leadership Awards LA1-08014 (J.P.C.) and DISC2-09624 (J.P.C.), NHLBI grant U54-HL-119893 (J.P.C.), and the Tower Cancer Research Foundation Career Development Grant (M.R.).

## Author contributions

J.P.C. and M.E.J. conceived of and designed the study. Y.Z. and M.R. performed the majority of the experiments and analyzed the data. Y.Z. and M.R. designed experiments. E.D., H.J.G. and M.E.J. designed and performed organic synthesis. M.Q., C.M.T. and H.A.H. performed experiments and analyzed data. X.Y., J.K., L.Z., T.F., M.L., K.P. and R.D. performed experiments. J.W. and W.M. contributed to the design and interpretation of the studies, and contributed to writing the paper. Y.Z., M.R., M.E.J. and J.P.C. wrote the paper.

## Additional information

**Competing interests:** Authors M.R., M.E.J. and J.P.C. are inventors on patent applications US 16/304,427 and EP 17803701.6, and PCT application US 2018/063,074, which are relevant to the use of PTPσ inhibitors described here. All other authors declare no competing interests.

