## [Peer Review File · Nature Communications]

Reviewers' comments:

Reviewer #1 (Remarks to the Author):

The paper by Zhang et al. describes a small molecule inhibitor of protein tyrosine phosphatase σ which the authors claim promotes regeneration of hematopoietic stem cells (HSCs). The authors show various data in both mouse and human systems suggesting an effect (direct or indirect) on the number of progenitors or HSCs using a variety of experimental systems, mainly post-. The data are convincing that the molecule has an effect on hematopoiesis, less clear is whether there is an effect on HSCs and on HSC proliferation (as opposed to survival). Mechanistic analysis links the effect to BCL-XL and Mcl1 downstream of the small GTPase Rac via modulation of p250GAP. The latter data would also favor an effect on survival pathways rather than proliferation pathways and some of these supporting data are somewhat weak. In addition, an overall criticism of the work is that some of the data are statistically significant but modest with a large overlap between experimental and control groups. Finally, in many cases it is not clear how the authors decided on the specifics of the model systems (eg timing of analysis after irradiation, doses of irradiation, dosage of inhibitor used).

Major comments

1. There is little data in the paper that actually supports the contention that the effect of this inhibitor is on HSC proliferation. Most data simply show an effect on apoptosis and biologic assay show post-irradiation maintenance of HSC function or immunophenotyped. This is also more consistent with the demonstrated role of Rac in HSCs (Gu et al., Science, 2003). Perhaps the effect is more on the RhoA or CDC42 pathway?
2. Other than PLA, is there direct biochemical evidence that PTP σ interacts with p250GAP and that this leads to a biochemical alteration in p250GAP function. Who many cells were used to generate the data in Figure 3A, where there is significant overlap between the experimental group and controls.
3. P250GAP is generally considered a GAP for RhoA and CDC42. The authors reference the result of yeast-two-hybrid screen. In this regard, it would useful to show directly the effect on Rac activity via a PBD pull down assay. In this regard, Rac is known as essential for HSC/P migration and adhesion and hyperactivation of Rac has been associated with increased adhesion. Did the authors investigate the effects of DJ001 on cell shape, adhesion and migration, all of which are critical to HSC function in the marrow cavity?
4. In multiple figures (eg 3d and 3g; 3f) it is not clear what the dots in the figures represent (Y axis labeled as %cells).
5. The authors utilize a large number of models without explanation. In some cases, cells are irradiated, in some cases not. In some cases animals are irradiated and in some cases not. It is not clear how these conditions were arrived at or why they were utilized. In this regard, are the PK or PD studies showing that concentrations reached in vivo are in the range needed as determined by in vitro assays?
6. In general the effects appears more related to post-irradiation modulation, which therefore suggests that the effects are not direct, but rather may be related secondarily to the complex response (cytokines and chemokines) post-irradiation. For example, in supplemental Figure 2, there appears to be no effect on hematopoiesis of 30 days of treatment in vivo of compound.
7. All the myelosuppression condition used in this manuscript was done by irradiations, have the authors tested the role of PTP σ inhibition in HSC regeneration under other myeloduppression condition, such as 5-FU treatment?
8. There appears to be some discrepancy between some data. For example in Figure 3d, viable cells after DJ001 treatment are ~45% while in Figure 3g, which appears to be done under the same condition, this is 20-25%. This difference is larger than the differences in the experimental group vs control group in each analysis.
9. While the authors claim an effect on multi-lineage progenitors/HSC, the data really supports a

different interpretation. For instance in Figure 2i, the largest effect is on CD3⁺ cells and in Figure 4f, the largest effect is on CD19⁺ cells, both cells of the lymphocyte lineage. Panel 2H the lower CD45.2⁺ cells in DJ001-treated cell reconstitution at 4 weeks suggests an effect on short-term progenitor engraftment. Is there a difference in progenitor phenotype in the BM of treated mice?

10. For the survival study, how many times was the analysis performed and was the survival sustained after 30 days?

11. The authors utilize a second compound DJ009 to “confirm our therapeutic development strategy” without explaining what is meant. This should be added.

12. Did the authors demonstrate a difference in Rac activation in Ptpns^{-/-} vs WT mice at baseline as would be predicted?

13. Panel 3C & F-I- An EHT1864 alone control should be included to ensure as a severe effect of Rac1 inhibition (in relation to vehicle and treatment groups) on survival, cell death, CFC formation, may change the interpretation of how DJ001 is impacting these processes in a Rac1-dependent manner. Other comments related to specific figures:

Figure 2:

A. The numbers of CFU-GEMM colonies shown in Figure 2a and 2b are extremely low. The authors should increase the number of LSK cells or WBM cells to start the CFU assay. The authors should also provide the colony numbers of CFU-G, CFU-GM and CFU-M in these two experiments.

B. Figure 2d, the authors should use WT mouse bone marrow cells as a control to show the gating strategy for the c-Kit⁺Sca-1⁺ cell population. Apparently, the Supplementary Figure 3C didn't show a distinctive c-Kit⁺/Sca-1⁺ cell population. Because the irradiation dramatically decreased the LSK cell number in the irradiated mouse BM, it is not clear how the authors defined the gate for c-Kit⁺/Sca-1⁺ cells based on a few LSK cells.

C. The transplantation experiment (Figure 2h) was done in a competitive setting. Are DJ001-treated irradiated mouse bone marrow cells able to reconstitute the hematopoiesis in a non-competitive setting?

Figure 3:

A. Figure 3c, the authors should provide the western blot data to show the p-PAK1 level.

B. Figure 3e, have these gene expression data been normalized to an internal control (such as Gapdh)?

Figure 4:

A. In Figure 4d, the decreased apoptotic human HSCs with the treatment of DJ001 is less dramatic as the decreased apoptotic mouse LSK cells with the treatment of DJ001 shown in Figure 3d. Does this correlate with the gene expression level change of Bcl-xL and Mcl1 genes? The author should also check the gene expression level of Bcl-xL and Mcl1 of human HSCs with or without treatment of DJ001.

B. In Figure 4e and 4f, the authors should also provide the data for the human CD34⁺CD38⁻ cells engraftment in the BM of NSG transplanted mice.

Reviewer #2 (Remarks to the Author):

Zhang et al. report in this manuscript that an inhibitor of the transmembrane protein tyrosine phosphatase-sigma promotes mouse and human hematopoietic recovery following DNA damage injury in both in vitro and in vivo settings. The authors propose that inhibition of PTP-sigma takes the effect by enhancing Rac1/2 signaling and upregulation of the anti-apoptotic protein BCL-xL. The hematological analyses were well designed and the manuscript was well written. However, the mechanistic analyses need to be strengthened. The drawback of this study is that the role of PTP-

sigma in hematopoietic cell regeneration has been published by the same group and really the inhibitor of this phosphatase showed only moderate effects.

Some specific comments: The function of the PTP-sigma inhibitor in hematopoietic cell recovery from irradiation may not take place only at the stem cell level. It is likely that the drug acts at multiple levels in the hematopoietic hierarchy. Authors should show the effects on stem cells and lineage progenitors and mature cells in the inhibitor-treated mice. Also, it is not clear why the authors think that the inhibitor functions only through Rac1/2 signaling. PTP-sigma might be involved in multiple signaling pathways. Does the inhibitor still function in Rac1/2 knockout cells? Similarly, hyperactivation of Rac1/2 can cause multiple cellular effects, in addition to increased cell survival. It is not clear why BCL-xL is considered as the main downstream effector. Was the better recovery in the inhibitor-treated animals only associated with the upregulation of BCL-xL? Does the inhibitor still function in BCL-xL knockout cells? Were other downstream signaling pathways enhanced, in addition to Erk? Would it be possible that the inhibitor also functions by increasing cell proliferation?

Point-by-Point Response to Reviewers' Comments

Reviewer 1

Reviewer #1 (Remarks to the Author):

The paper by Zhang et al. describes a small molecule inhibitor of protein tyrosine phosphatase σ which the authors claim promotes regeneration of hematopoietic stem cells (HSCs). The authors show various data in both mouse and human systems suggesting an affect (direct or indirect) on the number of progenitors or HSCs using a variety of experimental systems, mainly post-. The data are convincing that the molecule has an effect on hematopoiesis, less clear is whether there is an affect on HSCs and on HSC proliferation (as opposed to survival). Mechanistic analysis links the effect to BCL-XL and Mcl1 downstream of the small GTPase Rac via modulation of p250GAP. The latter data would also favor an affect on survival pathways rather than proliferation pathways and some of these supporting data are somewhat weak. In addition, an overall criticism of the work is that some of the data are statistically significant but modest with a large overlap between experimental and control groups. Finally, in many cases it is not clear how the authors decided on the specifics of the model systems (eg timing of analysis after irradiation, doses of irradiation, dosage of inhibitor used).

We appreciate the Reviewer's comments. Regarding the effect of DJ001 treatment on HSCs, we performed additional competitive repopulation assays that demonstrated that DJ001 treatment of irradiated mice increased the recovery of BM cells capable of multilineage repopulation in competitively transplanted, congenic mice at 20 weeks post-transplantation. Recipient mice transplanted with BM cells from irradiated, DJ001-treated mice displayed significantly increased donor myeloid, B cell and T cell percentages in the peripheral blood compared to mice transplanted with irradiated, saline-treated control mice (**Figure 2h**). These data support the conclusion that DJ001 treatment promoted the recovery of BM HSCs with multilineage repopulating capacity following total body irradiation.

Regarding the effects of DJ001 treatment on HSC proliferation after irradiation, we have performed additional experiments that showed that DJ001 treatment increased the percentage of BM ckit+sca-1+lin- (KSL) stem/progenitor cells in G2/S/M phase at 36 hours compared to control BM KSL cells (**Figure 4a**). We show further that DJ001-mediated increase in BM KSL proliferation is associated with upregulation of Cdk2 expression in KSL cells and that DJ001-mediated induction of Cdk2 expression in BM KSL cells can be suppressed by shRNA – mediated silencing of Rac1 (**Figure 4b,c**). These data suggest that DJ001 promotes HSC recovery via increased hematopoietic stem/progenitor proliferation as well as the anti-apoptotic effects shown in **Figure 3**.

In order to address the Reviewer's concern regarding some modest effects with overlap between the comparison groups, we repeated several critical experiments, including the measurements of peripheral blood complete blood counts in irradiated, DJ001-treated mice (**Figure 2c**), bone marrow KSL cells and ckit+sca-1-progenitor cells (**Figure 2e**), mouse survival studies following total body irradiation (**Figure 2f**), the 20 week

competitive repopulation assays to confirm an effect of DJ001 treatment on HSCs with multilineage repopulating capacity (**Figure 2g**), and analysis of BM KSL cell apoptosis (**Figure 3e**). These repeat analyses have confirmed our original findings, strengthened the statistical analyses of differences between the groups and increased the evident magnitude of effects of DJ001 on hematopoietic regeneration.

We appreciate the Reviewer's concern regarding the rationale for the different experimental conditions that we have tested. We have provided additional information in the revised manuscript to clarify the rationale for the different radiation doses and culture conditions. For in vivo studies of hematopoietic regeneration, we irradiated mice with 750 cGy total body irradiation, which depletes HSCs and progenitor cells, causes pancytopenia and is lethal to approximately 50% of C57BL/6 mice. We also added a 5FU chemotherapy mouse model in response to the Reviewer's suggestion to determine the effects of DJ001 treatment on chemotherapy – induced myelosuppression (**Figure 2i**). For in vitro studies and mechanistic experiments, BM KSL cells were irradiated with 300 cGy, which is myelotoxic and allows measurement of recovery of BM progenitor cells in culture. We also evaluated the effects of DJ001 treatment on non-irradiated adult C57BL/6 mice in order to distinguish hematopoietic regenerative effects of DJ001 from effects on steady-state hematopoiesis.

Major Comments

1. There is little data in the paper that actually supports the contention that the effect of this inhibitor is on HSC proliferation. Most data simply show an effect on apoptosis and biologic assay show post-irradiation maintenance of HSC function or immunophenotyped. This is also more consistent with the demonstrated role of Rac in HSCs (Gu et al., Science, 2003). Perhaps the effect is more on the RhoA or CDC42 pathway?

We appreciate this concern. In the original manuscript, we focused on characterizing the effect of PTP σ inhibition on HSC survival after irradiation, as well as the functional effects of PTP σ inhibition on hematopoietic regeneration. In response to the reviewer's concern, we have performed more extensive analysis of the effects of PTP σ inhibition on HSC proliferation following irradiation. Following irradiation, DJ001 treatment increased the percentages of BM ckit+sca-1+lin- (KSL) stem/progenitor cells in G2/S/M phase compared to control, in association with increased phosphorylation of Cyclin dependent kinase 2 (revised **Figure 4**). Rac inhibition or inhibition of Cyclin dependent kinase 2 abrogated DJ001 – mediated induction of HSC proliferation, suggesting that this effect of DJ001 on HSC proliferation following irradiation was dependent on Rac pathway activation and Cyclin dependent kinase 2. These findings are not inconsistent with those of Gu et al., *Science* 2003, who reported that Rac1 – deficient HSPCs displayed decreased entry into G2/S/M phase in response to SCF.

We thank the Reviewer for the recommendation to interrogate the effect of DJ001 on RhoA and Cdc42. In response, we tested the effect of DJ001 treatment on the levels of RhoA-GTP and Cdc42-GTP in BM lin- progenitor cells. DJ001 treatment did not alter the levels of the activated forms of RhoA or Cdc42 in BM lin- progenitor cells (**Supplementary Figure 6**). Therefore, our data suggest that the majority of the effects of DJ001 on HSPCs are mediated through Rac1 activation, rather than through activation of RhoA or Cdc42.

2. Other than PLA, is there direct biochemical evidence that PTP σ interacts with p250GAP and that this leads to a biochemical alteration in p250GAP function. Who many cells were used to generate the data in Figure 3A, where there is significant overlap between the experimental group and controls.

We appreciate the recommendation to provide more evidence beyond the proximity ligation assay that DJ001 affects PTP σ and its interaction with p250GAP. In response, we developed a p250GAP phospho-tyrosine sandwich ELISA to allow measurement of p250GAP phosphorylation in the presence and absence of DJ001 (revised **Supplementary Figure 6a**). DJ001 treatment for 10 minutes significantly increased p250GAP phosphorylation in BM lin⁻ progenitor cells compared to control cultures, consistent with biochemical alteration of p250GAP in response to DJ001 (revised **Figure 3a**). For the proximity ligation assay in the original submission, each data point represented the number of PTP σ – p250GAP complexes (red dots) per KSL cell per field of view. Ten fields of view were evaluated per group. In order to improve the clarity of data presentation, we have revised this Figure panel (revised **Figure 3b**) to show the number of PTP σ – p250GAP complexes per KSL cell in each condition (n=48 cells in the control group; n=36 cells in the DJ001 treatment group). The numbers of PTP σ – p250GAP complexes were significantly decreased in KSL cells treated with DJ001 ($p < 0.001$).

3. P250GAP is generally considered a GAP for RhoA and CDC42. The authors reference the result of yeast-two-hybrid screen. In this regard, it would be useful to show directly the effect on Rac activity via a PBD pull down assay. In this regard, Rac is known as essential for HSC/P migration and adhesion and hyperactivation of Rac has been associated with increased adhesion. Did the authors investigate the effects of DJ001 on cell shape, adhesion and migration, all of which are critical to HSC function in the marrow cavity?

We appreciate the concern that DJ001 might be modulating HSPC functions via regulation of p250GAP and RhoA and Cdc42. In response and as noted above, we directly measured RhoA-GTP and Cdc42-GFP levels in BM lin⁻ progenitor cells in response to DJ001 treatment and observed no change in RhoA-GTP or Cdc42-GFP levels in response to DJ001. We attempted to perform p250GAP and RAC1 pulldown assay in multiple cell types (primary murine BM cells, mouse 3T3 cells and human HEK 293T cells), but were unable to detect specific protein bands of corresponding interaction partners. p250GAP is a large protein (230 kD) and may interact only transiently with RAC1 to regulate phosphorylation. Additional troubleshooting over several months will be required for us to make this pull down assay feasible, so we have focused on responding to other concerns in this review that were feasible to address in our laboratory.

We thank the Reviewer for the question regarding the effects of DJ001 treatment on the migration and adhesion of hematopoietic stem and progenitor cells. Since our primary focus in this study was to evaluate endogenous HSC and progenitor cell regeneration in response to myelosuppressive injury, we focused our mechanistic studies on DJ001 effects on HSC survival and proliferation as shown in revised **Figure 3** and **Figure 4**. However, we very much appreciate the role of Rac proteins in regulating HSC and progenitor cell homing, localization and retention in the BM (Gu et al., *Science* 2003; Cancelas et al., *Nat Med* 2005; Mulloy et al., *Blood* 2010). In response, we evaluated the effects of DJ001 treatment on BM KSL cell migration through endothelial cell monolayers (Liu et al., *Methods Mol Biol* 2011). We observed no effect of DJ001 treatment on BM KSL cell migration through EC monolayers in vitro (revised **Supplementary Figure 7a**). We also observed no effect of

DJ001 treatment on the homing of transplanted sca-1+lin- cells to the BM at 18 hours post-transplant (revised **Supplementary Figure 7b**). These results do not exclude the possibility that systemic administration of DJ001 might promote in vivo hematopoietic regeneration via Rac1 activation and augmented HSC and progenitor cell adhesion and localization within supportive BM vascular and perivascular niches. We agree with the Reviewer that further study of the effects of DJ001 treatment on HSC localization and retention within the BM niche and the role of Rac1 in this process will be important to pursue, but we respectfully submit that these studies will require substantial additional imaging and microscopic analyses over time in both non-irradiated and irradiated mice, under multiple treatment conditions. We plan to pursue these questions in subsequent studies.

4. In multiple figures (eg 3d and 3g; 3f) it is not clear what the dots in the figures represent (Y axis labeled as %cells).

We apologize for the confusion created by the labeling of the y axes in these specific Figures. In revised **Figure 3e**, we have simplified the presentation to only show the comparison of percentages of Annexin+7AAD- apoptotic cells and have removed the plot of “live” cells as these results are redundant to the analysis of apoptotic KSL cells. The y axis of revised Figure 3 now reads “%Apoptotic KSL cells” to more accurately describe the population of apoptotic cells being shown. We have also added detail to the Figure Legend to clarify what populations are being compared in revised **Figure 3e**.

5. The authors utilize a large number of models without explanation. In some cases, cells are irradiated, in some cases not. In some cases animals are irradiated and in some cases not. It is not clear how these conditions were arrived at or why they were utilized. In this regard, are the PK or PD studies showing that concentrations reached in vivo are in the range needed as determined by in vitro assays?

We thank the Reviewer for bringing the concern regarding the rationale for the different experimental conditions to our attention. In response, we have provided more information in the text as to the basis for the different in vitro and in vivo studies with and without radiation. Our main objective was to determine the effects of PTP σ inhibition on hematopoietic regeneration following clinically relevant myelosuppression, using total body irradiation (TBI) and 5FU chemotherapy mouse models to assess these effects. We also evaluated the effects of DJ001 treatment on non-irradiated BM KSL cells in vitro and in non-irradiated mice in order to understand the effects of PTP σ inhibition on the hematopoiesis during homeostasis.

We have performed PK studies in mice following single subcutaneous dosing of 5 mg/kg DJ001 in adult C57BL/6 mice and these studies showed that the concentration of DJ001 increased to 100 ng/ml in the peripheral blood within 1 hour after subcutaneous administration and levels of approximately 10 ng/ml were observed through + 24 hours following administration (revised **Supplementary Figure 4f**). Our in vitro studies showed that concentrations of 10 – 100 ng/ml DJ001 significantly increased the number of BM CFCs in 3 day culture of BM KSL

cells (**Supplementary Figure 3b**). These results suggested that subcutaneous administration of 5 mg/kg DJ001 achieved sufficient concentrations in vivo for bioactivity.

6. In general the effects appears more related to post-irradiation modulation, which therefore suggests that the effects are not direct, but rather may be related secondarily to the complex response (cytokines and chemokines) post-irradiation. For example, in supplemental Figure 2, there appears to be no effect on hematopoiesis of 30 days of treatment in vivo of compound.

We agree that our in vivo studies suggest significant effects of systemic administration of DJ001 on hematopoietic regeneration following myelosuppressive irradiation, whereas systemic administration of DJ001 to non-irradiated mice had no major effects at 30 days. We appreciate that DJ001 administration could be mediating effects on HSC regeneration and hematopoietic reconstitution after TBI via indirect effects on the inflammatory response or the microenvironment. However, in the in vitro setting, in which there were no microenvironment cells or microenvironment – derived inflammatory cascade involved, DJ001 treatment induced RAC pathway activation and BCL-XL expression in irradiated BM KSL cells (**Figure 3c,d** and **Figure 3g,h**), promoted the recovery of CFCs from irradiated BM KSL cells (**Figure 3i**) and induced BM KSL cell cycle progression at 36 hours following irradiation (**Figure 4a**). Furthermore, DJ001 treatment of irradiated human CD34+ stem/progenitor cells, in the absence of other supportive cells, inhibited human CD34+ cell apoptosis (**Figure 5c**), induced BCL-XL expression in CD34+ cells (**Figure 5d**) and promoted the regeneration of human HSCs capable of repopulating NSG mice (**Figure 5f,g**). These studies suggest that at least some of the in vivo effects of DJ001 administration were mediated by direct effects on BM HSPCs in myelosuppressed mice. We recognize that additional studies using conditional, cell-specific deletion of *PTPσ* in BM hematopoietic cells and niche cells, coupled with DJ001 administration, will be necessary to define the cellular targets of DJ001 in the setting of myelosuppression and we are working toward developing *PTPσ* – floxed mice for this purpose.

7. All the myelosuppression condition used in this manuscript was done by irradiations, have the authors tested the role of PTPσ inhibition in HSC regeneration under other myeloduppression condition, such as 5-FU treatment?

We thank the reviewer for this important suggestion to determine whether *PTPσ* inhibition could also promote hematopoietic regeneration in mice following 5FU chemotherapy. Given the large numbers of patients that receive myelosuppressive chemotherapy regularly in the treatment of various cancers, this is a very important question to address. In response, we tested whether systemic treatment with DJ001 could promote hematologic recovery in C57BL/6 mice following 5FU chemotherapy administration. Interestingly, DJ001 treatment of 5FU – treated mice significantly accelerated the recovery of peripheral blood WBCs and neutrophils at day +14 compared to 5FU – treated mice that received vehicle (**Figure 2i**). These results suggest that systemic administration of a *PTPσ* inhibitor has therapeutic potential to accelerate hematologic recovery in patients who receive myelosuppressive

chemotherapy for cancer. We thank the Reviewer for this suggestion as these results have substantially increased the translational significance of our manuscript.

8. There appears to be some discrepancy between some data. For example in Figure 3d, viable cells after DJ001 treatment are ~45% while in Figure 3g, which appears to be done under the same condition, this is 20-25%. This difference is larger than the differences in the experimental group vs control group in each analysis.

We appreciate this concern. We have repeated these analyses and have confirmed the significant differences in the percentages of apoptotic BM KSL cells in the control, irradiated mice and irradiated, DJ001 – treated mice at 24 hours following 500 cGy TBI. These results are shown in the revised **Figure 3e**.

9. While the authors claim an effect on multi-lineage progenitors/HSC, the data really supports a different interpretation. For instance in Figure 2i, the largest effect is on CD3+ cells and in Figure 4f, the largest effect is on CD19+ cells, both cells of the lymphocyte lineage. Panel 2H the lower CD45.2+ cells in DJ001-treated cell reconstitution at 4 weeks suggests an effect on short-term progenitor engraftment. Is there a difference in progenitor phenotype in the BM of treated mice?

We appreciate this important question as to the effect of DJ001 treatment on BM progenitor cell populations and short-term repopulating cells in light of the 4 week engraftment data. In response, we analyzed DJ001 – treated and control mice at day +10 following TBI and found no differences in percentages of common myeloid progenitors (CMPs), granulocyte monocyte progenitors (GMPs) or myeloid erythroid progenitor cells (MEPs), and a small increase in the percentage of common lymphoid progenitors (CLPs) in DJ001 – treated mice at day +10 following TBI (**Supplementary Figure 4d**). It remains possible that DJ001 treatment could affect the functional capacity of short-term repopulating cells in vivo and this will be the subject of future studies in our laboratory.

Regarding the donor cell engraftment within specific lineage populations in the competitive transplantation studies, we have completed additional competitive repopulation assays and have also revised our analysis approach to simply show the absolute percentages of total donor CD45.2+ cells, donor CD45.2+Mac1/Gr1+ myeloid cells, donor CD45.2+B220+ B cells and donor CD45.2+CD3+ T cells within the PB of recipient mice at 20 weeks post-transplant. As shown in **Figure 2h**, recipient mice transplanted with BM cells from irradiated, DJ001 – treated donors displayed increased percentages of total CD45.2+ cells and increased percentages of CD45.2+Mac1/Gr1+ myeloid cells, CD45.2+B220+ B cells and CD45.2+CD3+ T cells compared to mice transplanted with an equal dose of BM cells from irradiated, vehicle – treated mice. We believe this analytical approach to measuring donor cell lineage engraftment provides more clear insight into the effect of DJ001 treatment on donor HSC function, compared to the analytical method we utilized in the original submission in which we showed the percentages of different lineages within the engrafted donor CD45.2+ cells.

Regarding the engraftment of human hematopoietic cells in NSG mice in revised **Figure 5g**, we note that, in addition to the increase in human B cell engraftment, we observed a significant increase in human CD45+CD33+ myeloid engraftment and human CD45+CD3+ T cell engraftment at 12 weeks in NSG mice transplanted with DJ001 – treated human CD34+ cells compared to mice transplanted with control – treated human CD34+ cells. Taken together, our murine and human repopulating assays support the conclusion that DJ001 treatment increased the recovery of hematopoietic stem/progenitor cells with multilineage repopulating capacity in vivo.

10. For the survival study, how many times was the analysis performed and was the survival sustained after 30 days?

In response to this concern, we repeated the survival study. The combined survival results, shown in revised **Figure 2f**, confirmed the strong effect of DJ001 treatment on mice survival following lethal dose TBI. All mice were euthanized at day +40 in both groups. We now show the survival curves through day +40 in revised **Figure 2f**.

11. The authors utilize a second compound DJ009 to “confirm our therapeutic development strategy” without explaining what is meant. This should be added.

We apologize for the lack of clarity in this statement. We were trying to convey that we have a broader strategy to synthesize several candidate PTP σ inhibitors for the purpose of identifying a lead compound for clinical translation. DJ009 is a structural analogue of DJ001 that we have synthesized which demonstrates strong PTP σ inhibitory activity and comparable bioactivity toward promoting hematopoietic regeneration in irradiated mice. We have edited the manuscript text to simply state that our results with DJ009 provide additional evidence that PTP σ inhibition promotes hematopoietic regeneration.

12. Did the authors demonstrate a difference in Rac activation in Ptp σ ^{-/-} vs WT mice at baseline as would be predicted?

Yes. As shown in revised **Figure 3c**, BM KSL cells from PTP σ ^{-/-} mice displayed increased Rac activation at baseline compared to BM KSL cells from PTP σ ^{+/+} mice.

13. Panel 3C & F-I- An EHT1864 alone control should be included to ensure as a severe effect of Rac1 inhibition (in relation to vehicle and treatment groups) on survival, cell death, CFC formation, may change the interpretation of how DJ001 is impacting these processes in a Rac1-dependent manner.

We appreciate this concern and have repeated these experiments to include the EHT1864 alone treatment arm to compare with control cultures, DJ001 alone, and DJ001 + EHT1864. These results are shown in revised **Figure 3e** and **Supplementary Figure 6d**.

Other comments related to specific figures:

Figure 2:

A. The numbers of CFU-GEMM colonies shown in Figure 2a and 2b are extremely low. The authors should increase the number of LSK cells or WBM cells to start the CFU assay. The authors should also provide the colony numbers of CFU-G, CFU-GM and CFU-M in these two experiments.

We thank the reviewer for bringing this to our attention. In response, we have repeated the assays and show the numbers of CFU-GM, BFU-E and CFU-GEMM for each study in revised **Figure 2a** and **Figure 2b**.

B. Figure 2d, the authors should use WT mouse bone marrow cells as a control to show the gating strategy for for the c-Kit+Sca-1+ cell population. Apparently, the Supplementary Figure 3C didn't show a distinctive c-Kit+/Sca-1+ cell population. Because the irradiation dramatically decreased the LSK cell number in the irradiated mouse BM, it is not clear how the authors defined the gate for c-Kit+/Sca-1+ cells based on a few LSK cells.

We thank the Reviewer for this suggestion and we have added the flow cytometric gating for BM ckit+sca-1+lin- cells from non-irradiated C57BL/6 mice, as well as the isotype control gate to revised **Supplementary Figure 4b**. During the hematopoietic regeneration phase (day +10 to day +17) following TBI, we have found that the flow cytometric profile of BM ckit+sca-1+lin- cells changes compared to homeostasis, such that ckit surface expression is more dim during the early hematopoietic recovery period (Doan P, et al. *Nature Medicine* 2013;19:295-304; Himgurg H, et al. *Nature Medicine* 2017;23:91-99). We have made the same observation in the current study and have adjusted our flow cytometric gate for BM ckit+sca-1+lin- cells to include ckit+ bright and ckit+ dim cells. We have adjusted our calculations of BM ckit+sca-1+lin- cells and ckit+sca-1-lin- progenitor cells based on the adjusted gating strategy shown in revised **Figure 2d**.

C. The transplantation experiment (Figure 2h) was done in a competitive setting. Are DJ001-treated irradiated mouse bone marrow cells able to reconstitute the hematopoiesis in a non-competitive setting?

We appreciate this question. We have not yet completed non-competitive BM transplant assays using DJ001-treated, irradiated BM cells, but this will be pursued in future studies. In this revision, we sought to focus our additional experiments on addressing the concerns raised by the Reviewer above and the concerns raised by Reviewer 2.

Figure 3:

A. Figure 3c, the authors should provide the western blot data to show the p-PAK1 level.

B. Figure 3e, have these gene expression data been normalized to an internal control (such as Gapdh)?

We utilized flow cytometry to measure phospho-PAK in BM KSL cells treated with and without DJ001. We agree that Western blot would be an ideal way to measure phospho-PAK1 in response to DJ001, but we have not found it to be feasible to complete Western blot analysis for phospho-proteins in primary hematopoietic stem cell populations, which are quite low in number. In order to interrogate signaling mechanisms in primary hematopoietic stem/progenitor cell populations, we have utilized flow cytometric analysis when reagents are available for such studies.

We used the *delta delta Ct* method to quantify the expression of target genes relative to Gapdh expression, as described by Livak K et al., *Methods* 25:402-408, 2001. The detailed method is provided in the revised Methods section.

Figure 4:

A. In Figure 4d, the decreased apoptotic human HSCs with the treatment of DJ001 is less dramatic as the decreased apoptotic mouse LSK cells with the treatment of DJ001 shown in Figure 3d. Does this correlate with the gene expression level change of Bcl-xL and Mcl1 genes? The author should also check the gene expression level of Bcl-xL and Mcl1 of human HSCs with or without treatment of DJ001.

We appreciate this suggestion and, in response, we measured the gene expression of *BCL2L1* and *MCL1* in human CD34+CD38- cells with and without DJ001 treatment. As shown in revised **Figure 5d**, DJ001 treatment strongly induced *BCL2L1* and *MCL1* expression in irradiated, human CD34+CD38- cells. Upon repeating these experiments several times, we have observed that the decrease in the level of apoptosis in DJ001-treated human HSPCs is comparable to that observed in DJ001-treated, irradiated, murine KSL cells (**Figure 3e** and **Figure 5c**). The changes in gene expression of *BCL2L1* and *MCL1* in human CD34+CD38- cells were also comparable to the magnitude of increase observed in these genes in irradiated, DJ001-treated murine KSL cells.

B. In Figure 4e and 4f, the authors should also provide the data for the human CD34+CD38- cells engraftment in the BM of NSG transplanted mice.

We thank the Reviewer for this suggestion. We cannot show data for human CD34+CD38- cell engraftment in NSG mice because we did not measure CD38 expression in the BM of NSG mice transplanted with human CD34+ cells.

Reviewer 2

Zhang et al. report in this manuscript that an inhibitor of the transmembrane protein tyrosine phosphatase-sigma

promotes mouse and human hematopoietic recovery following DNA damage injury in both in vitro and in vivo settings. The authors propose that inhibition of PTP-sigma takes the effect by enhancing Rac1/2 signaling and upregulation of the anti-apoptotic protein BCL-xL. The hematological analyses were well designed and the manuscript was well written. However, the mechanistic analyses need to be strengthened. The drawback of this study is that the role of PTP-sigma in hematopoietic cell regeneration has been published by the same group and really the inhibitor of this phosphatase showed only moderate effects.

We appreciate the Reviewer's comments and wish to emphasize two points in response. First, the development of deliverable pharmacologic approaches to promote HSC regeneration or hematopoietic reconstitution following myelosuppressive chemotherapy or irradiation has proven to be very challenging for the field despite several decades of research. More broadly, the development of phosphatase inhibitors for any therapeutic application has proven to be enormously challenging. As such, our description of a novel small molecule, allosteric inhibitor with specificity for PTP σ that can be systemically administered, promotes hematopoietic regeneration in mice following chemotherapy or TBI, while also driving regeneration of human HSCs capable of engrafting NSG mice, represents a significant contribution on several fronts. Secondly, in response to the Reviewer's concern regarding the magnitude of the effects of DJ001 treatment, we have repeated several aspects of our analysis, including the analysis of peripheral blood WBC and neutrophil recovery, BM KSL recovery, competitive repopulation assays and the mice survival study with and without DJ001 administration and have validated our original findings throughout. We highlight as well that DJ001 treatment also promoted a potent recovery of human HSCs following irradiation, as measured by several – fold increased multilineage engraftment in NSG mice at 12 weeks following transplantation, suggesting high translational significance. We believe that this magnitude of hematopoietic response provides the basis for therapeutic potential.

Some specific comments: The function of the PTP-sigma inhibitor in hematopoietic cell recovery from irradiation may not take place only at the stem cell level. It is likely that the drug acts at multiple levels in the hematopoietic hierarchy. Authors should show the effects on stem cells and lineage progenitors and matures cells in the inhibitor-treated mice.

We thank the Reviewer for this important suggestion. In response, we have measured the levels of Mac1/Gr1+ myeloid cells, B220+ B cells and CD3+ T cells in the PB of irradiated mice treated with and without DJ001. We found that DJ001 did not alter the percentages of any of these mature cells subsets at day +10 after TBI (revised **Supplementary Figure 4e**). We also measured the percentages of BM progenitor cell subsets at day +10 after TBI and treatment with or without DJ001. As shown in **Supplementary Figure 4d**, DJ001 treatment did not alter the percentages of BM CMPs, GMPs, or MEPs, and was associated with a small increase in CLPs.

Also, it is not clear why the authors think that the inhibitor functions only through Rac1/2 signaling. PTP-sigma might be involved in multiple signaling pathways. Does the inhibitor still function in Rac1/2 knockout cells?

We appreciate this concern and agree that other pathways may be involved besides RAC1 or RAC2 signaling pathways. In response to the Reviewer's specific question, we developed a viral shRNA targeting *Rac1* and tested whether DJ001 treatment would promote hematopoietic progenitor cell recovery following irradiation in the setting of *Rac1* silencing. As shown in revised **Figure 3i**, DJ001 treatment did not promote BM CFC recovery following irradiation in BM KSL cells treated with *Rac1* shRNA. These data strongly suggest that DJ001 – mediated hematopoietic stem/progenitor cell recovery is dependent on RAC1 pathway activation.

Similarly, hyperactivation of Rac1/2 can cause multiple cellular effects, in addition to increased cell survival. It is not clear why BCL-xL is considered as the main downstream effector. Was the better recovery in the inhibitor-treated animals only associated with the upregulation of BCL-xL? Does the inhibitor still function in BCL-xL knockout cells?

We agree that other downstream effectors besides BCL-XL certainly contribute to the beneficial effects of DJ001 treatment on HSC and progenitor cell recovery following myelosuppression. In revised **Figure 4**, we show that DJ001 also increased the proliferation of BM KSL cells following irradiation, associated with RAC pathway – dependent induction of Cdk2. Cdk2 inhibition blocked the increase in HSC proliferation in response to DJ001, suggesting that DJ001 – mediated induction of HSC proliferation was dependent on Cdk2.

We appreciated the reviewer's suggestion to test whether DJ001 – mediated BM hematopoietic progenitor cell recovery was dependent on BCL-XL. We developed a viral shRNA targeting *Bcl2l1*, which encodes BCL-XL, and found that DJ001 – mediated hematopoietic progenitor cell recovery following irradiation was abrogated in BM KSL cells treated with *Bcl2l1* shRNA (revised **Figure 3i**). These results suggested that DJ001 – mediated hematopoietic progenitor cell recovery was dependent, at least in part, on induction of BCL-XL.

Were other downstream signaling pathways enhanced, in addition to Erk? Would it be possible that the inhibitor also functions by increasing cell proliferation?

We appreciate this question as to whether other cellular mechanisms, such as HSC proliferation, were affected by DJ001. Additional experiments have demonstrated that DJ001 treatment increased the percentages of BM KSL cells in G2/S/M phase of cell cycle early (+36 hours) following irradiation compared to irradiated, control KSL cells (revised **Figure 4**). This increase in HSC proliferation following irradiation and DJ001 treatment was associated with increased *Cdk2* expression in BM KSL cells, and inhibition of Cdk2 suppressed DJ001 – mediated HSC proliferation following irradiation. Suppression of *Rac1* expression via pre-treatment of BM KSL cells with *Rac1* shRNA blocked DJ001 – mediated induction of *Cdk2* expression, suggesting that DJ001 – mediated induction of *Cdk2* expression was dependent on Rac1 activation (revised **Figure 4c**). We have modified the Discussion to state that other signaling pathways are likely activated in HSCs in response to DJ001, whereas our results highlight the importance of BCL-XL and Cdk2 activities, downstream of Rac pathway activation, in response to DJ001.

In summary, we appreciate the thorough reviews and specific recommendations from the Reviewers toward improving the manuscript. We have attempted to address all of the major concerns raised by both Reviewers with

additional experiments and more definitive analyses, in keeping with their guidance. We believe the revised manuscript has been significantly improved as a result.

We thank you for your consideration of the revised manuscript and hope for a favorable review.

Best regards,

Reviewers' comments:

Reviewer #1 (Remarks to the Author):

The authors of this manuscript performed extensive followup experiments regarding the reviewers' comments that has solidified some of the current data and interpretations. Overall the manuscript reads well, has clarity, and communicates the work of the authors. However, some clarity is still required regarding the mechanism of DJ001 in vitro versus in vivo at homeostasis and after myelosuppression.

For instance, it is still not clear that the effects of DJ001 are necessarily directly on HSC or secondarily via radiation-induced cytokines. The authors propose that because in vitro experiments are performed in the absence of . cells this rules out inflammatory effects; however, progenitors have the capacity to produce copious amounts of cytokines under stress conditions (Zhoa et al., Cell Stem Cell 2014).

Moreover, the difference in in vitro and in vivo results at homeostasis still raise some confusion about the precise action of DJ001 that has not been clarified in the current manuscript.

Although the authors performed additional competitive repopulation assays and demonstrated that DJ001 treatment of irradiated mice increased the recovery of BM cells capable of multilineage repopulation in the peripheral blood, how the DJ001 treatment directly affects the HSC frequency and proliferation in transplanted mice in vivo is still not clear.

The in vitro cell cycle analysis data shown in Figure 4a is clear, however the cell cycle analysis using BM cells isolated from transplanted mice with DJ001 treatment would provide more convincing information.

The overall data provided from this manuscript still favor the effect of DJ001 on HSC survival rather than on HSC proliferation.

Reviewer #3 (Remarks to the Author):

I only really feel comfortable dealing with the 'small molecule synthesis' part of this paper. Others will be better placed to review the cell biology.

The synthetic chemistry here is rather incremental but well done.

The beginning is a little thin on detail. Exactly how did the authors leap from the initial structure they used to the (only 2) compounds identified? This seems a rather large leap of faith. "Structural comparison" is very vague - what were the parameters used to identify their similarity? It is appreciated that this is not the main thrust of the article, but to narrow down a lead structure to just two structural analogues from a library of 80,000 is a significant fishing exercise.' Aside from this the synthetic portion of the paper (which is not very much) is fine. The methods of synthesis are good and the product analysis is fine. Their demonstration that it is a non-competitive inhibitor and modelling into a putative allosteric binding site is convincing.

Reviewer #4 (Remarks to the Author):

This is a revised manuscript from Zhang et al examining the role of PTP-sigma in regulating hematopoietic stem cell regeneration in both mice and humans after radiation and chemotherapy induced stress. Several concerns were raised during the initial review of this manuscript which authors

have addressed by conducting additional experiments. The manuscript is profoundly improved and is of high quality and significant in the field of stem cell regeneration.

Point-by-Point Responses to Reviewers

We appreciated the positive comments from all the Reviewers and have responded to the remaining concerns of Reviewer 1 and Reviewer 3, including additional analyses suggested by Reviewer 1. We believe the manuscript has been substantially improved as a result. Below, we provide point-by-point responses with Reviewers' comments shown in *italics*:

Reviewer 1

The authors of this manuscript performed extensive followup experiments regarding the reviewers' comments that has solidified some of the current data and interpretations. Overall the manuscript reads well, has clarity, and communicates the work of the authors. However, some clarity is still required regarding the mechanism of DJ001 in vitro versus in vivo at homeostasis and after myelosuppression.

For instance, it is still not clear that the effects of DJ001 are necessarily directly on HSC or secondarily via radiation-induced cytokines. The authors propose that because in vitro experiments are performed in the absence of cells this rules out inflammatory effects; however, progenitors have the capacity to produce copious amounts of cytokines under stress conditions (Zhao et al., Cell Stem Cell 2014).

We appreciate this concern and the interesting results from Zhao et al., *Cell Stem Cell* 2014, which demonstrated that ckit⁺sca-1⁺lin⁻ (KSL) hematopoietic progenitors can produce substantial amounts of cytokines in response to inflammatory stresses. We have added this reference to our Discussion to provide context for our results. In order to address this question directly in our model, we have measured the levels of 38 different cytokines in 12 hour culture of non-irradiated BM KSL cells treated with and without DJ001, and in culture of irradiated KSL cells (300 cGy) treated with and without DJ001 (time point of analysis in keeping with Zhao et al. *Cell Stem Cell* 2014). As shown in revised **Supplementary Figure 6a and 6b**, at 12 hours following 300 cGy irradiation, KSL cell cultures contained modest increased concentrations of LPS-induced CXC chemokine (LIX), macrophage inflammatory protein – 1 alpha (MIP-1 α) and MIP-1 β compared to non-irradiated KSL cell cultures; no other cytokine concentrations were altered significantly by irradiation in vitro. DJ001 treatment had no effect on cytokine levels in cultures of non-irradiated BM KSL cells (**Supplementary Figure 6a and 6b**). Following 300 cGy, DJ001 treatment was associated with a significantly lower level of LIX, but no other significant changes were observed in other cytokine/chemokine levels by 2-way ANOVA.

Moreover, the difference in in vitro and in vivo results at homeostasis still raise some confusion about the precise action of DJ001 that has not been clarified in the current manuscript.

We appreciate this concern and agree that we observed differences in the effects of DJ001 in vitro on cultured BM CD34⁺KSL cells (increased KSL cell expansion and CFC output) compared to a lack of effects of systemic treatment with DJ001 on BM KSL cells and hematopoiesis in non-irradiated mice. One possible explanation for these results is a difference in concentrations of DJ001 achieved in culture conditions in vitro compared to that achieved in the BM following subcutaneous injections every Monday-Wednesday-Friday in our mice homeostasis studies. In our studies of irradiated mice, we administered DJ001 daily for 10 days, which likely achieved higher, more consistent levels of DJ001 in the BM during the treatment period. We have added comment in the Discussion to include this possible explanation. In the future, with additional resources, we will formally address the optimal dosage and schedule of DJ001 to administer to non-irradiated mice to determine effects on steady-state hematopoiesis.

We agree with the Reviewer that another explanation for the differences between DJ001 effects in vitro on BM KSL cells and following in vivo administration in non-irradiated mice would be indirect effects of DJ001 on cytokine levels in vivo. In order to address this directly, we measured a panel of

38 cytokines in the BM of non-irradiated C57BL/6 mice treated with DJ001 or vehicle for 5 days. As shown in **Supplementary Figure 6c and 6d**, we observed no effect of DJ001 on any cytokine levels in non-irradiated mice.

Total body irradiation (TBI) significantly increased the levels of SCF and pleiotrophin (PTN) and decreased levels of LIX and CXCL12 in the BM at day +5 compared to non-irradiated mice; DJ001 treatment was associated with an increase in CXCL12 and decreased PTN levels in irradiated mice (**Supplementary Figure 6c and 6d**). Taken together, these data confirm that TBI alters the concentrations of some HSC-supportive cytokines in the BM, whereas DJ001 treatment altered the concentrations of two growth factors in irradiated mice. Determination of the functional relevance of these changes will require further study.

Although the authors performed additional competitive repopulation assays and demonstrated that DJ001 treatment of irradiated mice increased the recovery of BM cells capable of multilineage repopulation in the peripheral blood, how the DJ001 treatment directly affects the HSC frequency and proliferation in transplanted mice in vivo is still not clear.

One possible explanation for the increased HSC repopulating capacity in the transplanted mice is that DJ001 treatment increased the recovery of functional HSCs in irradiated donor mice compared to irradiated, vehicle-treated controls. In keeping with this, irradiated, DJ001-treated mice displayed 2-fold increased percentages and numbers of BM KSL stem/progenitor cells compared to irradiated, control mice (**Figure 2d, e**). Comparisons of percentages and numbers of BM SLAM KSL cells at day +10 in irradiated mice treated with and without DJ001 would also be helpful in this regard, but we observed very small numbers of events in the flow cytometric analysis of SLAM KSL cells at this time point, so have refrained from including this population in the analysis.

In order to clarify the effects of DJ001 treatment on HSC proliferation in vivo and following transplantation into mice, we have performed two additional analyses. First, we measured BrdU incorporation in BM KSL cells in adult C57BL/6 mice at day +10 following 750 cGy TBI, treated with and without DJ001. As shown in **Supplementary Figure 7i**, DJ001 – treated mice displayed an increased percentage of BrdU⁺ KSL cells in the BM at day +10 compared to vehicle-treated controls. Second, in order to determine if DJ001 treatment of irradiated donor mice affected HSC proliferative potential after competitive transplantation into recipient mice, we measured BrdU incorporation in donor CD45.2⁺ cells in recipient CD45.1⁺ mice at day +7 and day +21 following competitive transplantation. Of note, donor cells were harvested from the BM of CD45.2⁺ mice at day +10 following 750 cGy TBI and daily treatment with DJ001 or vehicle, and then transplanted into lethally irradiated CD45.1⁺ recipient mice, as described in Figure 2. As shown in **Supplementary Figure 7j**, we observed no differences in BrdU incorporation in donor CD45.2⁺ cells at day +7 or day +21 post-transplant in recipient mice based on whether they had been exposed to DJ001 or vehicle treatment in donor mice.

The in vitro cell cycle analysis data shown in Figure 4a is clear, however the cell cycle analysis using BM cells isolated from transplanted mice with DJ001 treatment would provide more convincing information.

As described above, we analyzed BrdU incorporation in donor CD45.2⁺ hematopoietic cells at day +7 and day +21 following competitive transplantation into CD45.1⁺ recipient mice. We detected no difference in BrdU incorporation in DJ001-treated or vehicle-treated donor CD45.2⁺ cell in vivo at day +7 or day +21 post-transplantation in recipient CD45.1⁺ mice (**Supplementary Figure 7j**).

The overall data provided from this manuscript still favor the effect of DJ001 on HSC survival rather than on HSC proliferation.

We agree with the Reviewer, particularly in light of the additional experiments we performed that showed a significant, but modest increase in BM KSL proliferation at day +10 following DJ001 treatment in vivo and no durable alteration in the cell cycle status of BM cells from DJ001-treated mice following transplantation into congenic recipient mice. We have added a specific comment addressing this point in the revised Discussion.

Reviewer 3

I only really feel comfortable dealing with the 'small molecule synthesis' part of this paper. Others will be better placed to review the cell biology.

The synthetic chemistry here is rather incremental but well done. The beginning is a little thin on detail. Exactly how did the authors leap from the initial structure they used to the (only 2) compounds identified? This seems a rather large leap of faith. "Structural comparison" is very vague - what were the parameters used to identify their similarity? It is appreciated that this is not the main thrust of the article, but to narrow down a lead structure to just two structural analogues from a library of 80,000 is a significant fishing exercise.'

We thank the Reviewer for this feedback and recognize that our description as to how we identified the lead small molecules for further study lacked sufficient detail. In response, we have revised the first paragraph of the Results section (entitled "Development of a selective, allosteric PTP σ inhibitor") to include the specific biochemical features, the 2-arylamino-1-arylpropanones, that we found in common between the original compound, 6545075, and the 2 compounds we selected, 6515205 and 5483071, from the 80,000 molecule library. We also revised the final paragraph of the Results section entitled, "PTP σ inhibition promotes hematopoietic regeneration following irradiation or chemotherapy," to clarify that we selected an additional compound, DJ009, for further in vivo testing from a group of more than 50 structural analogues of DJ001 that we synthesized based on its strong PTP σ inhibitory activity.

Aside from this the synthetic portion of the paper (which is not very much) is fine. The methods of synthesis are good and the product analysis is fine. Their demonstration that it is a non-competitive inhibitor and modelling into a putative allosteric binding site is convincing.

We thank the Reviewer for these positive comments.

Reviewer 4

This is a revised manuscript from Zhang et al examining the role of PTP-sigma in regulating hematopoietic stem cell regeneration in both mice and humans after radiation and chemotherapy induced stress. Several concerns were raised during the initial review of this manuscript which authors have addressed by conducting additional experiments. The manuscript is profoundly improved and is of high quality and significant in the field of stem cell regeneration.

We thank the Reviewer for these very positive comments about the manuscript and its significance to the field.

REVIEWERS' COMMENTS:

Reviewer #1 (Remarks to the Author):

The authors have addressed in the main part my comments. There are critical experiments as they note which still need to be performed. Indeed, the lack of clarity around in vivo mechanism still weakens the manuscript's impact. It is perplexing that so much work has been done with this molecule but that schedule and dosing has not yet been established that would allow a more informed comparison between in vitro and in vivo effects. This seems to be a key unresolved issue. The new data provided should perhaps be used to edit the abstract such that it reflects this uncertainty.

Response to Reviewer

Reviewer 1 comment: The authors have addressed in the main part my comments. There are critical experiments as they note which still need to be performed. Indeed, the lack of clarity around in vivo mechanism still weakens the manuscript's impact. It is perplexing that so much work has been done with this molecule but that schedule and dosing has not yet been established that would allow a more informed comparison between in vitro and in vivo effects. This seems to be a key unresolved issue. The new data provided should perhaps be used to edit the abstract such that it reflects this uncertainty.

We appreciate Reviewer 1's concern and have added the following statement to the Discussion section to reflect the remaining uncertainty from our cytokine analyses referred to by the Reviewer: "These results suggest that DJ001 promotes HSC regeneration primarily via direct effects; however, indirect effects of DJ001 via undiscovered mechanisms cannot be not excluded." I believe this statement is more appropriate in the Discussion section rather than in the Abstract since the Abstract word count is limited and should reflect the most significant findings in the paper. We will move to the statement to the Abstract if the Editor advises us to do so.